# Compartmentalized mitochondrial ferroptosis converges with optineurin-mediated mitophagy to impact airway epithelial cell phenotypes and asthma outcomes

Kazuhiro Yamada [1,2], Claudette St. Croix[3], Donna B. Stolz[3], Yulia Y. Tyurina[1], Vladimir A. Tyurin[1], Laura R. Bradley[1], Alexander A. Kapralov [1], Yanhan Deng[1,4], Xiuxia Zhou[1], Qi Wei [1], Bo Liao[1,5], Nobuhiko Fukuda[1,6], Mara Sullivan [3], John Trudeau[1], Anuradha Ray [7], Valerian E. Kagan [1], Jinming Zhao [1] ✉ & Sally E. Wenzel [1] ✉

A stable mitochondrial pool is crucial for healthy cell function and survival. Altered redox biology can adversely affect mitochondria through induction of a variety of cell death and survival pathways, yet the understanding of mitochondria and their dysfunction in primary human cells and in specific disease states, including asthma, is modest. Ferroptosis is traditionally considered an iron dependent, hydroperoxy-phospholipid executed process, which induces cytosolic and mitochondrial damage to drive programmed cell death. However, in this report we identify a lipoxygenase orchestrated, compartmentally-targeted ferroptosis-associated peroxidation process which occurs in a subpopulation of dysfunctional mitochondria, without promoting cell death. Rather, this mitochondrial peroxidation process tightly couples with PTEN-induced kinase (PINK)−1(PINK1)-Parkin-Optineurin mediated mitophagy in an effort to preserve the pool of functional mitochondria and prevent cell death. These combined peroxidation processes lead to altered epithelial cell phenotypes and loss of ciliated cells which associate with worsened asthma severity. Ferroptosis-targeted interventions of this process could preserve healthy mitochondria, reverse cell phenotypic changes and improve disease outcomes.

Disruption of mitochondria, the ATP-generating engines of life, leads to pathologic dysfunction and cell death. Ferroptosis, an iron-dependent lipid peroxidation-induced form of programmed cell death, has been consistently associated with mitochondrial abnormalities and dysfunction[1,2]. Yet the importance of the interactions and the directionality of the mitochondrial-ferroptotic effects remain poorly understood and are likely context specific[3]. Potential interactions range from induction of non-enzymatic ferroptosis, driven by internal mitochondrial redox imbalance to external disruption of mitochondrial membranes from lipoxygenase-linked ferroptosis.

15 lipoxygenase-1 (15LO1), an enzyme abundantly expressed in asthmatic human airway epithelial cells (HAECs), often in relation to Type-2 (IL-4/−13) immune processes and more severe disease, drives generation of these ferroptotic/oxidized phospholipids (PLox)[4–7]. When bound to phosphatidylethanolamine binding protein-1 (PEBP1), the preferred enzymatic substrate for 15LO1 changes from free arachidonic acid to arachidonoyl-containing - phospholipids, in particular phosphatidylethanolamine (PE)[5,8]. This switch then enables generation of 15 hydroperoxy-eicosatetraenoic acid-PE (15 HpETE-PE), which under homeostatic conditions, is neutralized by glutathione peroxidase-4 (GPX4) to its alcohol metabolite 15 hydroxy-eicosatetraenoic acid-PE (15 HETE-PE). While this process minimizes cell death and/or membrane damage, reduced glutathione (GSH) is oxidized in the process, also altering the intracellular redox balance[9,10].

Despite high 15LO1 expression in asthmatic airway epithelial cells, little cell death is observed. Recently, the *pro-survival* autophagy lipoprotein, microtubule associated light chain-3 (LC3-II), was observed to be increased in fresh asthmatic HAECs, in concordance with 15LO1 expression[11]. In vitro, as 15LO1-PEBP1 binding and ferroptotic activity increased, LC3 was no longer constrained by binding to PEBP1, but was available for lipidation and induction of autophagic membrane formation. This increase in LC3-II then limited ferroptotic (RSL3) induced cell death, perhaps explaining the limited cell death in vivo. This increased LC3-II also reduced the extracellular release of mitochondrial (mt), as opposed to double stranded nuclear DNA. While autophagy can lead to removal of multiple different cell organelles[12], this selective release of mtDNA suggested ferroptotically induced mitochondrial damage, driven by 15LO1-PEBP1 activity, with a protective mitophagic response. Although 15LO1 has previously been associated with mitochondrial loss and cell differentiation[13], the specific impact of lipoxygenase-induced ferroptosis targeted to mitochondria, the identification of a selective mitophagic response and the ultimate functional impact of this described convergence of cell death and survival pathways are unknown. We therefore hypothesized that 15LO1-PEBP1 activity programmatically drives a compartmentalized, enzymatic mitochondrial ferroptosis, which is structurally limited by activation of specific mitophagy pathways to both remove damaged mitochondria and preserve cell function. Using complementary studies of fresh asthmatic and cultured HAECs, we identify an *external* 15LO1, PLox and mitochondrial-targeted ferroptotic signal, an optineurin-driven, PTEN-induced kinase (PINK)−1/Parkin mediated mitophagic response and their combined functional relevance to ciliary cell phenotypes and worsened asthma outcomes.

## Results

### IL-13 induces mitochondria loss via 15LO1-dependent ferroptosis

Mitochondria drive cellular ATP formation, with damage often associated with disease. We previously reported that ferroptotic processes increased extracellular mtDNA release in HAECs stimulated with IL-13 and that LC3-II limited that increase[11]. Whether mitophagy is involved, the type of mitophagy executed and its intersection with ferroptosis, as well as the functional implications remain poorly understood. To address these questions, we utilized short (7 days) and longer term (14 days) HAEC cultures at air liquid interface (ALI), stimulated (or not) with IL-13 (10 ng/ml), beginning at day 0 or day 7 of ALI. At 7 or 14 days, cells were fixed and TOM20 or ATP Synthase (ATPsynthase) immunofluorescent (IF) imaging utilized to identify mitochondria. Mitochondria volume and size under both short (identified by TOM20) and longer (identified by ATPsynthase) term cultures were decreased under IL-13 conditions compared to time matched media controls (Fig. 1A and Supplementary Fig. 1A, B). Transmission electron microscopy (TEM) images identified a subciliary, apical positioning of mitochondria under control conditions (blue arrows), with IL-13 stimulation decreasing mitochondria density and abundance (yellow arrows) (Fig. 1B). Similarly, intracellular mtDNA (determined by PCR)

was lower under IL-13 as compared to control conditions (Fig. 1C). This loss of mitochondria occurred in the absence of greater cell death under IL-13 conditions (measured by LDH release) and without concurrent loss of nuclear DNA-associated material (histone levels by western blot (WB)) (Supplementary Fig. 1C). To begin to address whether this IL-13-induced loss of mitochondria was due to ferroptosis associated mitochondria damage, IL-13 (10 ng/ml) or control treated cells were stimulated (or not) with a brief (2 hrs) exposure to RSL3 (10 μM to both apical and basal media). Short RSL3 stimulation was chosen to only modestly initiate ferroptosis without execution of widespread cell death (and release of mtDNA). RSL3 stimulation under IL-13 conditions, compared to IL-13 alone, induced mitochondria swelling and damage by TEM (Fig. 1B, last panel). These effects on mitochondria were confirmed by IF/CF volume/sphericity analysis using TOM20 antibodies. As shown in Fig. 1D−F, 2 hrs of RSL3 stimulation did not reduce overall mitochondria volume (by TOM20 IF staining) compared to IL-13 alone. However, RSL3 stimulation further increased fragmentation of mitochondria (smaller size and increased sphericity), consistent with previous reports of mitochondrial damage in cells undergoing ferroptotic cell damage[3,14]. RSL3 treatment of control cells (with low 15LO1 expression) had little effect on mitochondria number, size and fragmentation/sphericity (Supplementary Fig. 1D−F), supporting the selectivity of these effects to conditions in which 15 LO1 is elevated.

To determine whether activation of the 15LO1 pathway contributes to ferroptotic loss of intact mitochondria, IL-13 + RSL3 stimulated cells were treated with the ferroptosis inhibitor FER-1 and the 15LO1 inhibitor BLX2477. As seen in Fig. 1G, last two panels, pretreatment with FER-1 or BLX2477 appeared to limit the mitochondrial damage. Protective effects of 15LO1 mediated ferroptosis inhibition were confirmed using siRNA knockdown (KD) of 15LO1 (siALOX15) with dicer siRNA (dsiRNA) under IL-13 conditions (7 days starting at day 0 of ALI). siALOX15 KD restored mitochondria volume, without impacting size as compared to control conditions (Fig. 1H and Supplementary Fig. 1G), while similarly increasing intracellular mtDNA levels (Supplementary Fig. 1H). These results suggest that 15LO1 pathway activity drives organelle targeted ferroptosis to damage and perhaps recycle mitochondria.

### 15LO1 is present in mitochondria and contributes to generation of ferroptotic lipids

15LO1-PEBP1 binding switches 15LO1's preferred substrate from free arachidonic acid (AA) to AA conjugated to PE (AA-PE), generating the hydroperoxy-phospholipid 15-hydroperoxyeicosatetraenoic acid−PE (15-HpETE-PE) to induce ferroptosis. To determine whether a mitochondrial associated 15LO1 was driving mitochondria loss under IL-13 conditions, we first determined whether 15LO1 colocalizes with mitochondria. Using ultracentrifugation with slight modification of the Mitochondria Isolation Kit (Abcam) manufacturer's instructions (see Methods), the mitochondrial, nuclear and cytosolic fractions were isolated under both control and IL-13 conditions for 7 days. After determining that the nuclear fraction included a mix of nuclear and cytoskeleton components with trace levels of mitochondria, it was discarded without further analysis (per manufacturer's instructions). Both cytosolic and mitochondrial fractions were analyzed by WB to confirm level of purity. The mitochondrial protein marker COXII was detected exclusively in the mitochondrial fraction and GAPDH was only detected in the cytosolic fraction (Fig. 2A). This relative purity was supported by measurement of compartment-specific lipids of interest, including cardiolipin (CL), phosphatidylserine (PS) (mitochondria vs cytosol specific) and bis(monoacylglycero)phosphatidic acid species (BMPs), as well as PE-plasmalogens (specific for lysosomes and peroxisomes). See Isolation of Mitochondria Fractions in Methods, and accompanying supplemental Fig. 8A−C for validation by specific lipids.

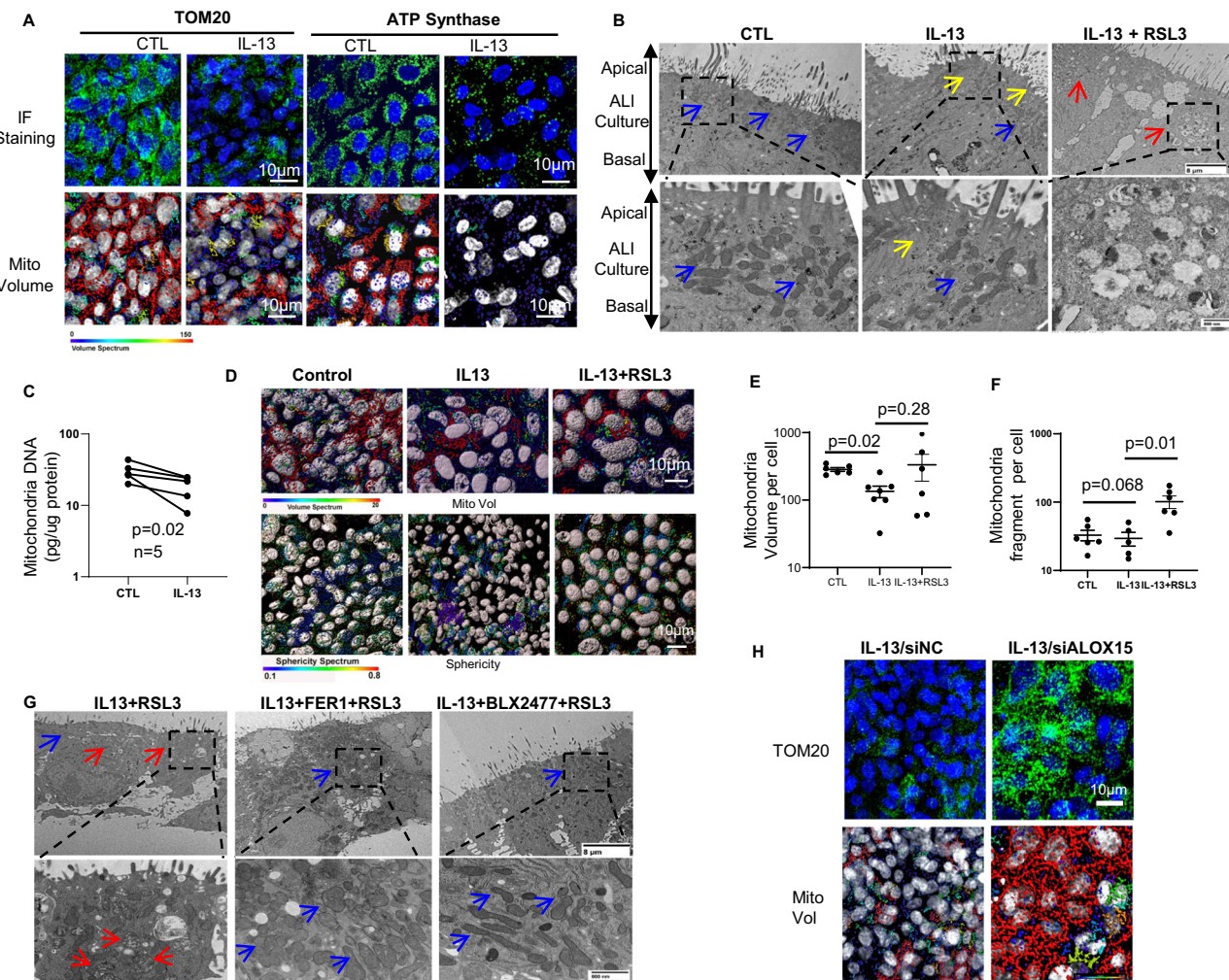

**Fig. 1 | IL-13 induces mitochondria loss via 15LO1-dependent compartmentalized ferroptosis. A** IL-13 decreases TOM20 and ATPsynthase staining (top panel; Green: TOM20 or ATPsynthase; Blue: DAPI) and measurements of mitochondria volume (bottom panel, the volume is pseudo-colored with red indicating the regions with the highest mitochondrial volume, as shown in the "*volume spectrum*") (Representative immunofluorescence (IF) staining-confocal (ICF) microscopy). Representative images from $n = 4$ biologic replicates (quantitative analysis in Supplementary Fig. 1A, B). **B** Mitochondria are enriched in apical/ciliary area and decreased by IL-13 under transmission electron microscopy (TEM). RSL3 induces mitochondrial swelling and damage (last panel). Blue arrows: high density normal mitochondria; Yellow arrows: low density mitochondria and loss. Red arrows: Swollen and damaged mitochondria (representative sample from multiple sections, $n = 1$). Scale bar, 8 μm (upper panel) and 800 nm (lower panel). **C** IL-13 decreases intracellular mitochondria DNA (paired T-testing), as measured by qPCR, normalized to total cell lysate protein. HAECs were stimulated with/without IL-13 for 7 days at day 0 of ALI. Paired T-testing of $n = 5$ biological replicates. **D** RSL3 induces mitochondrial fragmentation in IL-13-treated HAECS (IF/CF). HAECs stimulated with IL-13 for 7 days at Day 0 of ALI before RSL3 treatment (10 μM, 2 h). Cells fixed and stained with TOM20. **E** Mitochondria volume assessed using surface rendering of confocal z-stacks (Imaris, Bitplane) pseudo-colored as indicated by "*volume spectrum*" in (**D**) (top panel) by paired t-testing (**F**) mitochondria fragmentation assessed based on size and shape of mitochondria as reflected by sphericity in (**D**) (bottom panel, mitochondria are progressively smaller and rounder as color changes from purple to red). Data presented as mean ± s.d, and analyzed using unpaired T-testing based on $n = 5$–7 biological replicates. **G** Pretreatment of FER-1 and/or BLX2477 limit IL-13-induced mitochondrial damage. Red arrows: swollen damaged mitochondria. Blue arrows: high density and normal mitochondria (representative sample from multiple sections, $n = 1$). Scale bar 8 μm (upper panel) and 800 nm (lower panel). **H** siALOX15 transfection restores IL-13-induced mitochondria loss indicated by increases in mitochondrial volume (IF/CF). Representative images from $n = 4$ biologic replicates. Data quantification and RSL3 impact under control conditions in Supplementary Fig. 1G, H. Source data are provided as Source Data file.

15LO1 expression increased following IL-13 stimulation in both the cytosolic and mitochondrial fractions, with similar expression and subcellular location observed for glutathione peroxidase (GPX)4, the peroxidase critical for neutralization of ferroptotic hydroperoxy-phospholipids. PEPB1 was detected in both cytosolic and mitochondria fractions (Fig. 2A and Supplementary Fig. 2A–F). The co-localization of 15LO1 with mitochondria was confirmed by IF using 15LO1 and ATPsynthase staining followed by object based co-localization analysis as previously reported in ref. 5, with the region of overlap of the two emission profiles assessed using a binary 'having' statement (represented by yellow puncta in Fig. 2B). 15LO1/

ATPsynthase colocalization (Fig. 2B, top panel, and Supplementary Fig. 2G) as identified by common puncta (Fig. 2B, lower panel, and Supplementary Fig. 2H), was increased under IL-13 conditions.

To determine whether these in vitro observations were also seen ex vivo in freshly brushed asthmatic HAECs, similar IF staining was performed on epithelial brushing cytospins from both healthy control (HC) and asthmatic participants. As shown in Fig. 2C, high 15LO1 and low ATPsynthase staining was commonly observed in asthmatic as compared to healthy epithelial cells. ATPsynthase and 15LO1 colocalized (indicated in yellow) in apical regions of the cells, often just beneath the cilia (Fig. 2C).

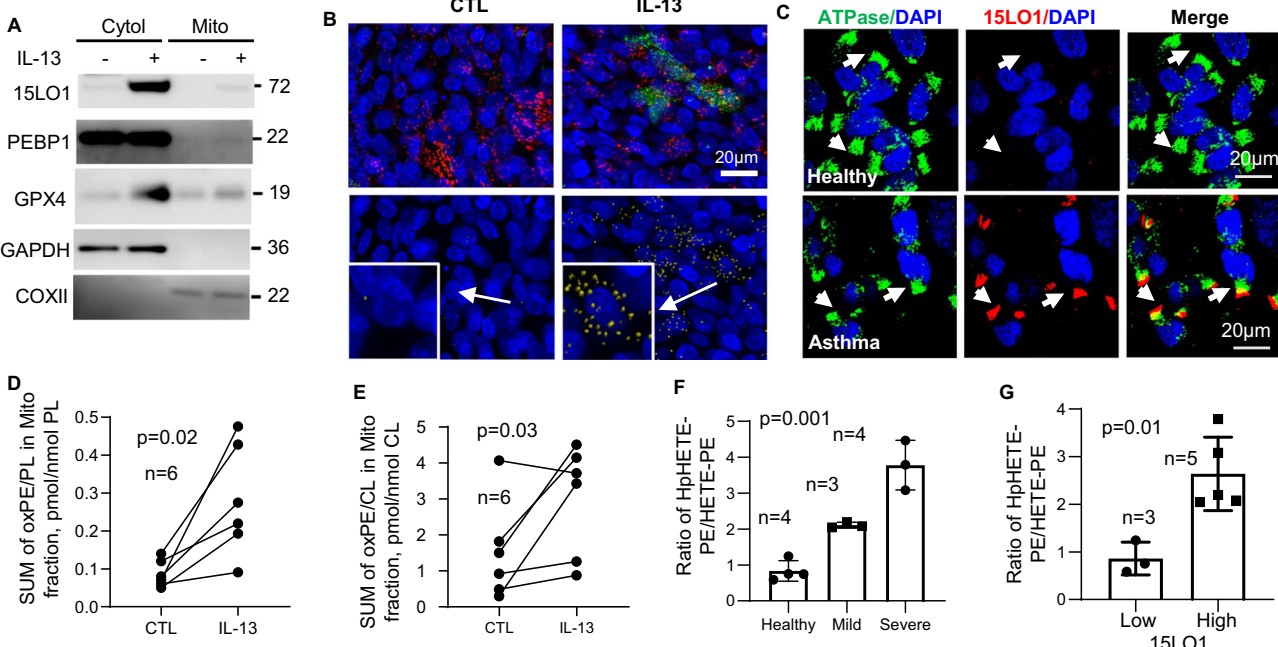

**Fig. 2 | Mitochondrial 15LO1 pathway activity drives compartmentalized ferroptosis. A** IL-13 induces ferroptosis-associated 15LO1 protein expression in the mitochondrial fractions. HAECs were stimulated with/without IL-13 for 7 days. Cytosolic and mitochondrial fractions were isolated by ultracentrifugation for WB analysis. Representative images from *n* = 3 biologic replicates, with data quantification analysis in Supplementary Fig. 2A–F. **B** IL-13 increases 15LO1/ATPsynthase colocalized puncta in HAECs in vitro. Top panel: IF/CF staining. Red: ATPsynthase; Green: 15LO1; Blue: DAPI; Yellow: colocalization. Bottom panel: co-localized 15LO1/ATPsynthase puncta in yellow. Lower left corner digitally magnified. Representative images from *n* = 3 biologic replicates, with quantitative analysis in Supplementary Fig. 2H. **C** 15LO1/ATPsynthase colocalization (IF/CF) in freshly blushed HAECs from healthy and asthmatic participants. Green: ATPsynthase; Red: 15LO1; Blue: DAPI;

Yellow: colocalization. White arrows: cilia area. Representative images from *n* = 3 biologic samples from individual donors in each group. **D** IL-13 increased oxPE in mitochondrial fractions in HAECs in vitro as normalized to the total intracellular phospholipid (PL) or (**E**) cardiolipin (CL) by LC/MS analysis. Significance analyzed by paired T-testing of *n* = 6 biological replicates. **F** The ratio of 15-HpETE-PE to 15 HETE-PE is higher in freshly brushed HAECs from severe asthmatic as compared to healthy control participants (one way ANOVA) (**G**) and higher in those participants (mild to severe asthma) with high 15LO1 protein (by WB). Data are presented as mean ± s.d, and differences determined by one-way ANOVA (**F**) and unpaired t-test (**G**) based on *n* = 3–5 biological samples from individual donors. Source data are provided as a Source Data file.

To directly measure 15LO1-generated ferroptotic lipids, HPLC/MS was utilized to detect both 15HpETE-PE and 15HETE-PE in mitochondrial fractions from HAECs as isolated previously (See Methods). Of note, 15HpETE-PE to 15 HETE-PE ratios were lower than previously reported in whole cell lysates[5], likely due to ongoing metabolism of 15 HpETE-PE to its alcohol over the prolonged fractionation processing time (about two hours longer than simple lysate analysis). Therefore, 15HpETE-PE and 15HETE-PE levels were combined to represent the total amount of 15LO1 products generated. Total oxidized PE (oxPE) (15 hydroperoxy + 15 hydroxy PE) was indexed to total phospholipids (PL) as well as to cardiolipin (CL), found exclusively in mitochondria[15–17]. IL-13 significantly increased total oxPE in mitochondria (Fig. 2D, E), confirming the presence of a localized ferroptotic signal in mitochondria.

Studies of isolated mitochondria are not feasible in freshly brushed airway cells, due to overall low cell numbers. However, to confirm similar relationships of 15LO1 expression to ferroptotic lipid levels ex vivo, 15 HpETE-PE and 15 HETE-PE were measured in freshly brushed HAECs ex vivo. Higher 15 HpETE-PE per PL was detected in fresh HAECs from asthmatic, as compared to HC HAECs (Fig S2I). However, unlike the fractionated cultured cells, the ratio of 15HpETE-PE/15HETE-PE favored the hydroperoxy form, with the highest ratios measured in severe asthma (SA) patients (Fig. 2F). To determine whether these levels associated with 15LO1, 15LO1 was measured by WB in cells from the same brushings ex vivo. The 15 HpETE-PE to 15 HETE-PE ratio was higher in HAECs with higher 15LO1 protein expression (Fig. 2G). These results are in line with ongoing 15LO1-driven ferroptotic activity in HAECs with high 15LO1 expression, with in vitro data supporting the presence and ferroptotic activity of 15LO1-PEBP1 in mitochondria.

## Mitochondrial membrane potential($\Delta\Psi_m$) loss and remodeling associate with ferroptosis

The mitochondrial dysfunction described above could also contribute to loss of $\Delta\Psi_m$, which is a known activator of mitophagy[18–21]. To determine whether IL-13 stimulated primary HAECs exhibit loss of $\Delta\Psi_m$, cells treated (or not) with IL-13 were evaluated by flow cytometry using the cell-permeant fluorescent dye, tetramethylrhodamine methyl ester (TMRM), that accumulates in mitochondria in a membrane potential–dependent way[22]. Two cell subsets were identified: Subset A having a higher level of TMRM fluorescence, and Subset B characterized by lower TMRM fluorescence (Fig. 3A). IL-13 stimulation decreased TMRM fluorescence compared to controls in both subsets, indicative of decreased $\Delta\Psi_m$ (Fig. 3A, B). Loss of $\Delta\Psi_m$ in vitro is supported ex vivo by measurements of CL, which is exclusively localized to inner mitochondrial membrane[17,23] and is thus readily available for peroxidation [(to oxidized CL (CLox)] by mitochondria generated reactive oxygen species. Interestingly, the amounts of CLox were very low in fresh primary HAECs ex vivo, without significant differences between healthy and asthmatic samples (Supplementary Fig. 3). Since CLox is effectively hydrolyzed by group VIB Ca2 + -independent phospholipase A2γ (iPLA2γ) yielding mono-lyso-CLs (mCLs) and oxygenated fatty acids as the major products[24,25] and iPLA2γ is predominantly distributed to mitochondria[26], mCLs levels were measured in fresh primary HAECs as well (See Table 1 for demographics of this subset). Figure 3C shows elevated levels of mCLs in Mild-Moderate (MOD) and Severe Asthma (SA) as compared to HC samples (Fig. 3C). Importantly, levels of these mCLs correlated strongly with forced expiratory volume in 1 second (FEV1)% predicted (*r* = 0.9, *p* < 0.0001)

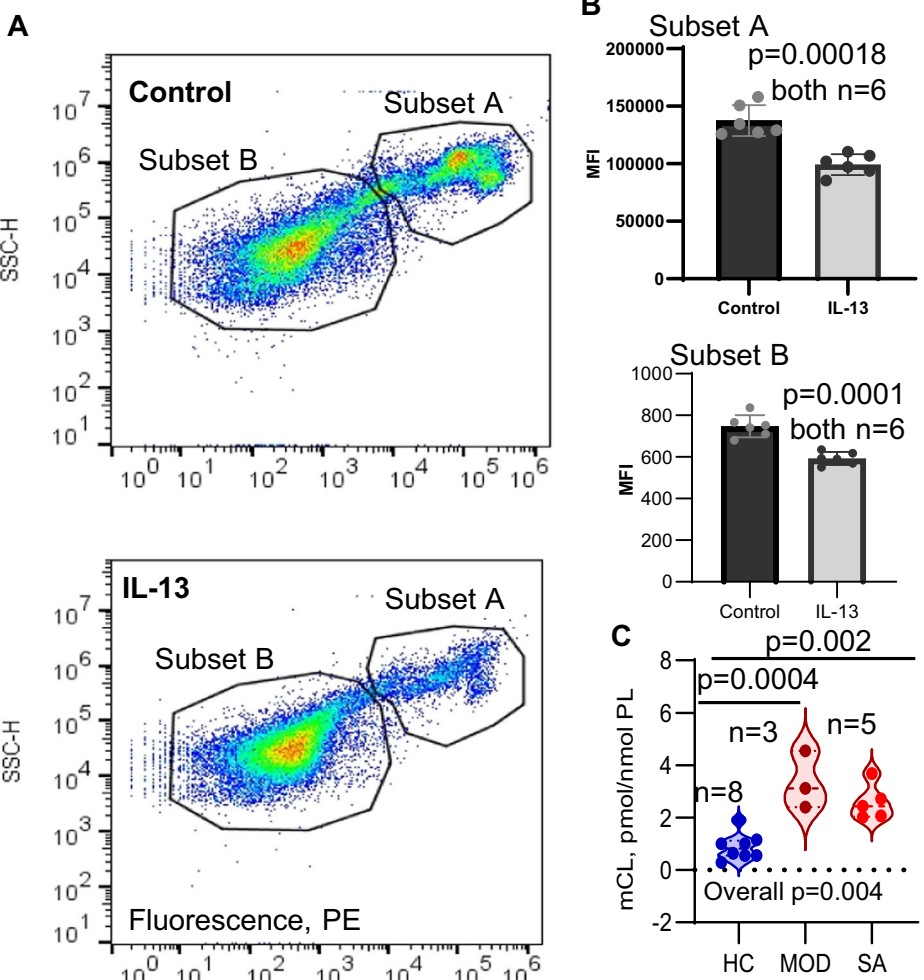

**Fig. 3 | IL-3 treatment decreases mitochondrial membrane potential in HAEC.** HAECs were cultured with media alone under ALI for 9 days and then stimulated or not with IL-13 for 5 days before harvested for membrane potential assessment by Tetramethylrhodamine, methyl ester (TMRM). **A** Scatter plot of side-scatter intensity versus TMRM fluorescence intensity for untreated (upper panel) and IL-13 treated cells (bottom panel). **B** The mean of fluorescence intensities (MFI) of subset A and subset B in control and IL-13 treated cells. Data are presented as mean ± s.d,

and significance analyzed by unpaired T-testing of $n = 6$ biological replicates. **C** LC/MS analysis showing elevated levels of mCLs in Mild-Moderate (MOD) and Severe Asthma (SA) as compared to HC samples (LC/MS analysis). Differences were determined by one-way ANOVA followed by intergroup unpaired t-testing of $n = 3$ to 8 biological samples per group from individual donors (see Table 1 for details). oxCL data in Supplementary Fig. 3. Source data are provided as a Source Data file.

and with fraction exhaled nitric oxide (FeNO), a marker of Type-2 inflammation (Spearman's Rho = 0.7, $p = 0.047$). There were no differences in mCL by gender. Women made up the majority of the participants as expected in an adult asthma cohort. Thus, mCL likely represents a remodeling of mitochondrial phospholipids, which may contribute to and be a consequence to mitochondrial dysfunction[27,28].

### Mitochondrial ferroptosis increases LC3 lipidation and mitophagy

Autophagy, a major evolutionarily conserved cell survival process, can also be selective, with autophagosomes forming around specific cargo, including mitochondria, in which case it is known as mitophagy[29]. We previously reported that 15LO1-PEBP1 dependent ferroptotic pathway activation co-dependently freed LC3 from PEBP1 for its lipidation/activation thereby upregulating compensatory autophagy in HAECs[11]. These combined processes lowered RSL3/ferroptosis mediated release of mtDNA and lessened cell death. To further understand the dynamic link of ferroptosis to LC3 activation, a time course for activation of these processes following IL-13 stimulation was performed (Fig. 4A). IL-13 rapidly (within 2 hrs) increased 15LO1 protein, preceding lipidation/

activation of LC3 by approximately 4 hrs, and thereby supporting the necessity of prior ferroptotic activity. 15LO1 and LC3 protein were also analyzed by IF after 5 days of IL-13 (Fig. 4B and Supplementary Fig. 4A). 15LO1 and LC3 co-localized under IL-13 conditions, particularly in basal cell types, consistent with our previous publication[30]. This co-localization was also observed in freshly brushed HAECs ex vivo. However, bronchoscopic brushings generally retrieve more superficial cell types than are observed in biopsies or ALI culture, with a high proportion of ciliated cells. 15LO1 and LC3 staining and co-localization was commonly observed in asthmatic cells, with 15LO1 and LC3 primarily co-localized in ciliated cells, particularly in their sub-ciliary region (Fig. 4C).

To determine whether this colocalization occurs in or near mitochondria, mitochondrial fractions were isolated as described earlier, and purification confirmed by specific protein analysis as before, but also including the mitochondrial proteins NDUFB8 and TOM20 (Fig. 4D). As we previously reported, IL-13 increased both LC3-I and LC3-II in cytosolic fractions (Fig. 4D, and Supplementary Fig. 4B), consistent with an increase in de novo synthesis as well as activation[11]. In contrast, only the active form of LC3, LC3-II, selectively and specifically increased in the mitochondrial fraction in response to IL-13, with

## Table 1 | Demographics of Participants with ex vivo Phospholipid Data

| | Healthy Controls | Mild-Moderate Asthma | Severe Asthma | Overall *p*-value |
|---|---|---|---|---|
| Age (mean±SD) | 42 ± 15 | 41±15 | 52 ± 13 | 0.46 |
| Gender (W/M) | 6/2 | 3/0 | 3/2 | 0.32 |
| Self identified race# | 1/9/1/0 | 2/9/0/0 | 1/8/1/0 | 0.72 |
| FEV1% pred (mean±SEM) | 114 ± 14 | 72 ± 12 | 78 ± 9 | 0.002 |
| FeNO (ppb) (mean±SD) | 17 ± 7 | 28 ± 11 | 40 ± 36 | 0.2 |
| Biologic Usage (Yes/No) ANOVA Analysis | N/A | N/A | 1/5 1-anti-IL-5 | |

*Black/White/Mixed/Asian.
*SD* Standard deviation, *SEM* Standard error of the mean, *IQR* Interquartile range.
*W/M* Woman/man, self reported.

inactive LC3-I exclusively found in the cytosolic fraction (Fig. 4C). HCQ pretreatment, which limits autophagic flux, further increased LC3-II accumulation in mitochondria fractions, supporting active autophagy (Fig. 4D 2$^{nd}$ panel, and Supplementary Fig. 4D). This exclusive accumulation of activated LC3-II in mitochondrial fractions after IL-13 simulation strongly indicates formation of mito-autophagic membranes (mitophagy), which was further supported by co-localization of LC3 with the mitochondrial protein ATPsynthase (Fig. 4E and Supplementary Fig. 4G, H). The yellow puncta indicate voxels where the two emissions are overlapping. Quantitative data in Supplementary Fig. 4H shows the number of co-localized puncta per cell and the total volume of overlapping voxels per cell. Both volume and puncta of LC3/ATPsynthase colocalization were increased after 7 days of IL-13. Consistent with Fig. 1A, B findings, reductions in mitochondrial marker proteins, NDUFB8 and COXII, were observed following IL-13 stimulation (Fig. 4D). These in vitro observations were again paralleled by LC3-ATPsynthase colocalization in freshly brushed HAECs (Fig. 4F). Similar to the 15LO1-ATPsynthase staining (Fig. 2C), LC3-ATPsynthase colocalized (indicated in yellow) in the same apical/subciliary region (Fig. 4F). Thus, high 15LO1/ferroptosis-associated activity converges with LC3-mitophagic processes in similar sub-apical (mitochondrial) regions in ciliary cells (ex vivo) as well as in basal cells from full thickness epithelial culture in vitro.

To confirm the importance of mitochondrial 15LO1/ferroptosis to LC3 activation and mitophagy, DsiRNA KD of 15LO1 (siALOX15) was performed, followed by IL-13 (7d) stimulation and isolation of mitochondrial and cytosolic fractions[6,31]. 15LO1 KD decreased both LC3-I and LC3-II in cytosolic fractions, but specifically decreased mitochondrial LC3-II accumulation (Fig. 4G and Supplementary Fig. 4E, F). 15LO1 KD also decreased ATPsynthase and LC3 co-localization by IF (Fig. 4H and Supplementary Fig. 4I-J). This link of mitophagy to 15LO1-driven ferroptosis was further supported by TEM which identified mitophagic double membranes under IL-13 + HCQ conditions (added to stabilize autophagic flux). Characteristic lipid bilayers engulfing mitochondria fragments indicative of mitophagy were observed only under IL-13/15LO1 high conditions (Fig. 4I, middle panel). Treatment with BLX2477 normalized the mitochondria with loss of mitophagic membranes. These results suggest that ferroptotic damage to mitochondrial membranes subsequently activates mitophagy.

### PINK/Parkin/Optineurin activation is critical for ferroptosis-linked mitophagy

Multiple factors and components have been reported to mediate mitochondria dysfunction induced mitophagy. To identify the mitophagic pathway activated under high 15LO1 conditions, we screened multiple previously identified proteins linked with mitophagy, including various receptor mediated pathways such as BNIP3, NIX, FUNDC1,and NDP52[32–35]. No consistent changes in receptor mediated pathways were observed, so they were not pursued further. Our current studies suggest ferroptosis induces loss of $\Delta\Psi_m$, a known activator of PINK1-Parkin associated mitophagy[36], and it has previously been reported to be activated in AECs[37–41]. In the present study, IL-13 induced expression of PINK1 protein, in association with increases in phospho-Parkin and decreases in total PARKIN in IL-13 stimulated HAECs (Fig. 5A and Supplementary Fig. 5A-E), supporting activation of PINK/Parkin driven mitophagy. Activation of PINK1-Parkin mitophagy requires a linking protein such as optineurin (OPTN) or NDP52 to bridge PINK1-Parkin to LC3 and subsequently initiate the elongation of the autophagosomal membrane[42,43]. OPTN expression markedly increased in response to IL-13 in whole cell lysates (Fig. 5A) and colocalized with LC3 by co-immunoprecipitation (Co-IP with LC3 as the pulldown antibody), with increased binding under IL-13 conditions (Fig. 5B). This increase in expression and binding was supported by co-localization of OPTN with LC3-II (IF/confocal) under IL-13 conditions (Fig. 5C and Supplementary Fig. 5F, G). To confirm that 15LO1 mediated ferroptotic processes drove PINK1-Parkin-OPTN executed mitophagy, the ferroptosis inhibitor FER-1 and the 15LO1 inhibitor BLX2477 were added simultaneous with IL-13. As shown in Fig. 5D and Supplementary Fig. 5H, I, both FER-1 and BLX2477 suppressed IL-13-induced OPTN and LC3-II expression. Additionally, BLX2477 decreased OPTN-LC3 colocalization (Fig. 5E and Supplementary Fig. 5J, K). These effects were further confirmed using 15LO1 KD (siALOX15) under IL-13/15LO1 high conditions. ALOX15 KD decreased expression of PINK1/pParkin, OPTN and LC3-II (Fig. 5F and Supplementary Fig. 5L–P), while decreasing the binding of OPTN with LC3 (Co-IP/WB) (Fig. 5F, bottom panel) and co-localization of OPTN with LC3-II (IF/confocal, Fig. 5G and Supplementary Fig. 5Q, R). Similar to our previous findings with LC3 KD[11], siOPTN KD enhanced RSL3-induced LDH release (Supplementary Fig. 5S) confirming its functional mitophagy associated protective effect on ferroptotic cells.

The biologic relevance of these mitophagic processes to human asthma was then evaluated in freshly brushed HAECs from asthmatic and healthy control (HC) participants ex vivo (Table 2 (Demographics for all WB clinical studies)). We previously reported increases in 15LO1 and LC3-II proteins by WB in relation to asthma severity[6,11,31]. Not surprisingly, OPTN protein by WB was also significantly higher in T2 high asthmatic HAECs (as defined by FeNO≥25 ppb) (Fig. 5H and Supplementary Fig. 5T), and positively correlated with 15LO1 (Spearman's rho = 0.49, *p* = 0.004), and LC3-II (rho = 0.38, *p* = 0.03), with marginally higher levels in women (*n* = 26) than men (*n* = 6) (*p* = 0.05). Both LC3-II and OPTN also correlated with FeNO [rho = 0.47, *p* = 0.007 (LC3-II) and rho = 0.36, *p* = 0.04 (OPTN)] n = 32, Table 2. IF/CF analysis confirmed colocalization of OPTN with LC3 (Fig. 5I), as well as with 15LO1 (Fig. 5J) in fresh asthmatic and to lesser degree in HC HAECs ex vivo. The apical/subciliary colocalization of OPTN with LC3 and/or 15LO1 paralleled the regions where 15LO1/ATPsynthase (Fig. 2C) and LC3/ATPsynthase (Fig. 4F) colocalized, potentially indicating the presence of active ferroptosis with mitophagic responses in some ciliated cells.

### The 15LO1 pathway links to ciliated cell phenotypic alterations and asthmatic outcomes

Ciliated airway epithelial cells generally make up the bulk of the airway epithelium where they drive mucociliary clearance. Primary ciliary dyskinesia which specifically impacts cilia function, is associated with severe airway dysfunction[44]. Asthmatic airways have been reported to have fewer ciliary cells (or expression of associated genes)[45–47], but whether the phenotype of the remaining ciliated cells differs from those found in HCs is unknown. As our data show abundant 15LO1 and

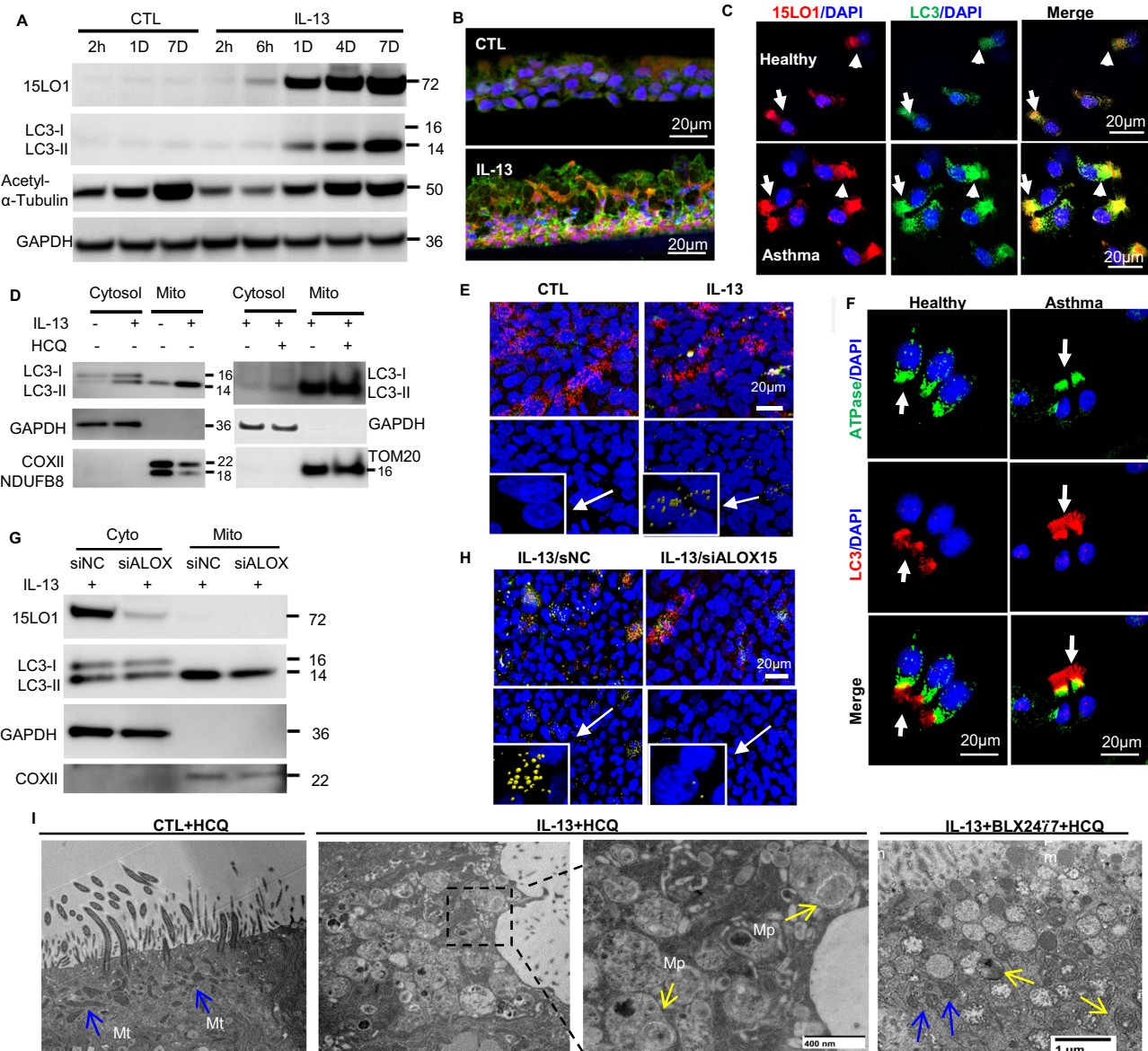

**Fig. 4 | Compartmentalized ferroptosis increases mitochondrial LC3 and mitophagy. A** IL-13-induces 15LO1 and LC3-II expression by WB over time, and 15LO1 increases prior to LC3-II. Representative image of $n = 3$ biological replicates. **B** IL-13 increased 15LO1/LC3 expression and colocalization in HAECs in vitro compared to control (CTL) (IF/CF on cross section of ALI membranes). Representative image of $n = 3$ biological replicates. **C** 15LO1 and LC3 colocalization in fresh healthy and asthmatic HAECs using IF/CF. Red: 15LO1; Green: LC3; Blue: DAPI; Yellow: colocalization. Representative images of $n = 3$ individual donors/group. **D** IL-13 induces LC3 expression with enrichment of activated LC3-II in mitochondrial fractions. HCQ (10 μM, overnight) pretreatment further increases LC3-II in mitochondria fractions (GAPDH as loading control). Representative images from $n = 3$ biologic replicates. **E** IL-13 increases LC3/ATPsynthase colocalization in HAECs in vitro. Top panel: IF staining, Red: ATPsynthase; Green: LC3; Blue: DAPI; Yellow: colocalization. Bottom panel: Colocalized LC3/ATPsynthase puncta indicated as yellow. Lower left corner: digitally magnified view of cells at tail of arrow. Representative images from $n = 3$ biologic replicates. **F** LC3/ATPsynthase colocalization in fresh HAECs using IF. Green: ATPsynthase; Red: LC3; Blue: DAPI; Yellow:

colocalization. White arrow: cilia area. Representative images of $n = 3$ individual donors/group. **G** 15LO1 KD (siALOX15) lowered mitochondrial LC3-II under IL-13 conditions. **H** 15LO1 KD/siALOX reduced LC3/ATPsynthase colocalization under IL-13 conditions. Top panel: IF staining, Red: ATPsynthase; Green: LC3; Blue: DAPI; Yellow: colocalization. Bottom panel: "Having" analysis of co-localized LC3/ATP-synthase puncta (indicated in yellow). Lower left corner: digitally magnified view of cells at tail of arrow. IF image scale bar: 20 μm. Representative images of $n = 3$ biologic replicates. **I** Mitophagic membranes are present in HAECs under IL-13 + HCQ conditions by TEM. The 15LO1 inhibitor BXL2477 (2 μM, overnight) lessens mitophagy and partially rescues mitochondria as compared to IL-13 + HCQ. HAECs cultured under ALI (9 days) before stimulation with IL-13 for 5 days), BLX2447 (2 μM, overnight), HCQ (10 μM) added 2 hrs before fixation. Blue arrows: intact mitochondria (Mt); yellow arrows/star: mitophagy (Mp) (representative sample from multiple sections, $n = 1$). Scale bar, 1 μm (regular panel) and 400 nm (enlarged panel). Data quantification and individual images by antibody in Supplementary Fig. 4. Source data provided as Source Data file.

associated mitophagic proteins in ciliated cells, the relationship of 15LO1 high conditions to cilia cell structure/phenotype was evaluated. Using scanning electron microscopy (SEM) of HAECs, stark loss of cilia numbers and alterations in their structure were observed under IL-13/15LO1 high, as compared to control conditions (Fig. 6A, top panel).

Using TEM, decreased and/or disrupted cilia (red arrows) and mitochondria loss (yellow arrow) were present under IL-13/15LO1 high conditions, as compared to more normal-appearing mitochondria (blue/green arrows) under control conditions (Fig. 6A, bottom panel). Mitochondria again were observed to be enriched (in both conditions)

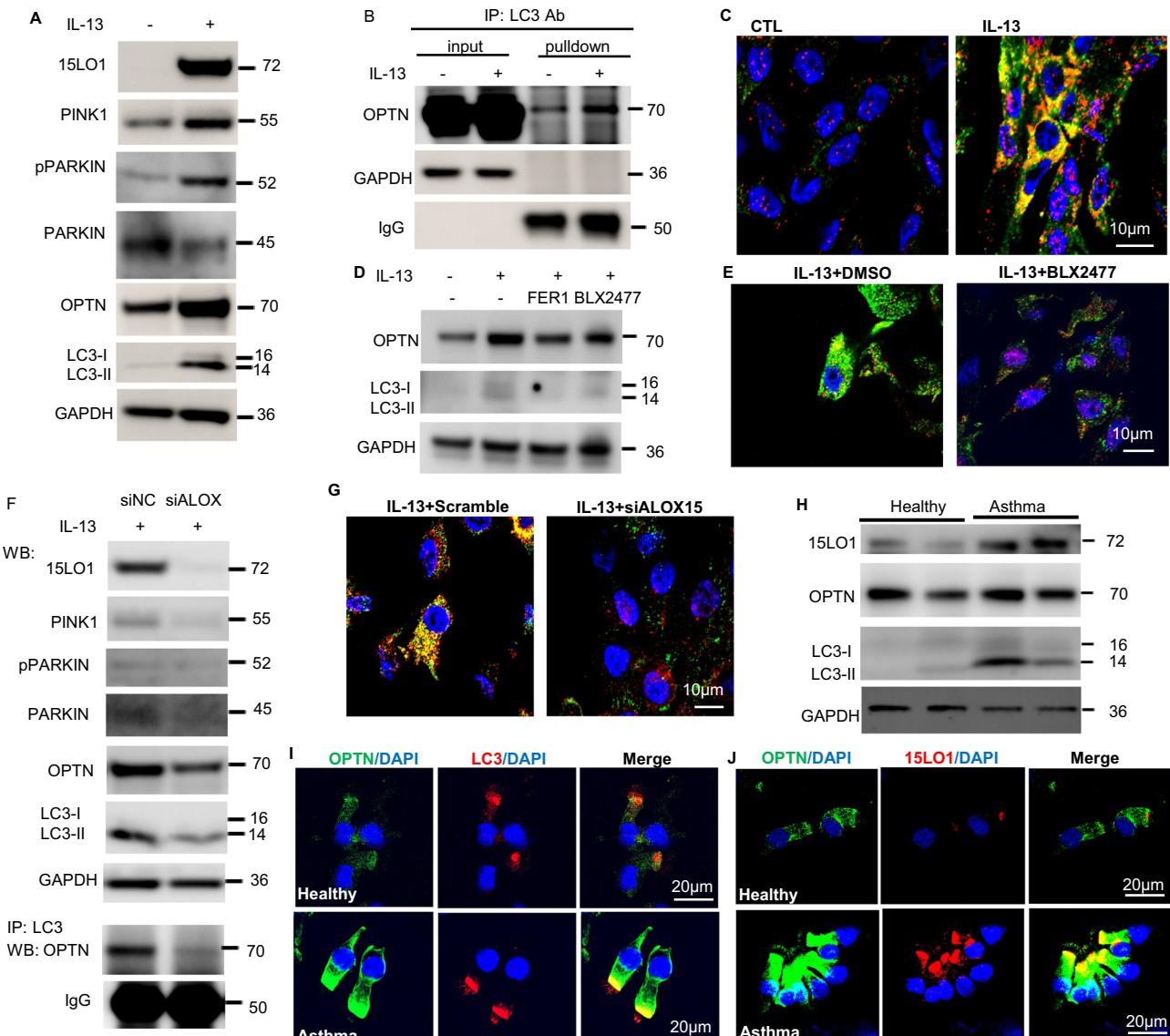

**Fig. 5 | 15LO1-dependent ferroptotic processes activate PINK/Parkin/Opti-neurin mitophagy. A** PINK1/pParkin/Optineurin (OPTN) /LC3-II are increased under IL13 conditions using Western Blot analysis (*n* = 3 biological replicates). **B** IL-13 increases OPTN co-immunoprecipitation with LC3 (Co-IP/WB) (*n* = 3 biological replicates). **C** Increased OPTN co-localization with LC3 under IL-13 conditions. IF/CF: Green: OPTN; Red: LC3; Blue: DAPI; Yellow: merge (*n* = 3 biological replicates). Scale bar as indicated. **D** Pretreatment with FER-1 and/or BLX2477 suppress IL-13-induced LC3-II ALI for 9 days and IL-13 for 5 days, with/without simultaneous FER-1 (1 μM) and/or BLX2477 (2 μM) each for days (*n* = 3 biological replicates). **E** 15LO1 inhibition with BLX2477 lowers LC3 colocalization with OPTN under IL-13 conditions. Green: OPTN; Red: LC3; Blue: DAPI; Yellow: merge. (*n* = 3 biological replicates). Scale bar as indicated. **F** 15LO1 KD/siLALOX15 lowers PINK1/PARKIN pathway activation (WB) and OPTN-LC3 binding (Co-PI/WB) under IL-13 conditions (*n* = 3 biological replicates), and (**G**) 15LO1/siLALOX15 lowers LC3 colocalization with OPTN under IL-13 conditions. Green: OPTN; Red: LC3; Blue: DAPI; Yellow: merge (*n* = 3 biological replicates). **H** Higher OPTN expression in association with 15LO1 and LC3 in freshly brushed asthmatic HAECs as compared to healthy controls by WB (*n* = 32 individual donors, see Table 2 and supp Fig S5T for densitometry quantification). **I** Representative IF/CF images showing OPTN-LC3 and (**J**) OPTN-15LO1 colocalization in freshly blushed HAECs from healthy and asthmatic parti-cipants (*n* = 3 individual donors/group). Green: OPTN; Red: LC3 or 15LO1; Blue: DAPI; merge. Source data provided as Source Data file.

in subciliary regions. Similarly, cells with fewer cilia and structural alterations (red arrows) as well as mitochondrial swelling (yellow arrow) were observed in freshly brushed asthmatic as compared to healthy HAECs (Fig. 6B, bottom panel at higher magnification). Inhibition of ferroptosis with the 15LO1 inhibitor FER-1 or BLX2477 restored tubulin expression by IF/CF (Fig. 6C/Supplementary Fig.6 A) and by WB (Fig. 6D and Supplementary Fig. 6B) and improved mature cilia development in 14 days of ALI cultures (Fig. 6C by SEM).

To determine whether some of these effects could be initiated at a very early stage, through effects of ferroptosis on basal cells, the progenitor cells required for all differentiated HAECs, an early ALI culture model, starting at Day 0 of ALI, was utilized where basal cells

predominate. IL-13 treatment starting at ALI day 0 almost completely inhibited cilia development, which was increased by BLX277 and FER-1 determined by IF/CF and SEM (Supplementary Fig. 6C–E). These results suggest that ferroptosis could initiate its effects at an early stage of cell differentiation.

Given the consistent findings of high 15LO1 protein in ciliated cells, typically in association with mitophagic markers, freshly brushed HAEC cytospins from 7 HCs and 31 asthmatic participants (Table 3), all with matching densitometrically quantified 15LO1 protein (by WB), were evaluated for cilia length, as observed on DiffQuik (*Thermo Fisher*) stained cytospins. Only cytospins with ≥75% of the cells identifiable as epithelial cells were included and a total of 100 epithelial cells were

**Table 2 | Demographics of Participants with ex vivo Western Blot Data ***

| | Healthy Controls | Mild-Moderate Asthma | Severe Asthma | Overall p-value |
|---|---|---|---|---|
| Age (mean±SD) | 40 ± 16 | 39 ±14 | 47 ± 12 | 0.36 |
| Gender (W/M) | 9/2 | 9/2 | 8/2 | 0.98 |
| Self identified race* | 1/9/1/0 | 2/9/0/0 | 1/8/1/0 | 0.72 |
| FEV1% pred (mean±SEM) | 103±13 | 88 ± 20 | 66 ± 22 | 0.0004 |
| FeNO (ppb) (mean±SD) | 13±5 | 31 ± 20 | 58 ± 13 | 0.001 |
| Biologic Usage (Yes/No) | N/A | N/A | 4/6 3-anti-IL-5, 1-anti-IL-4R. | |

ANOVA Analysis.
*8 participants overlap with Table 1.
#Black/White/Mixed/Asian.
SD Standard deviation, SEM Standard error of the mean, IQR Interquartile range.
W/M Woman/man, self reported.

counted in a blinded manner. Cilia length was qualitatively defined as short, medium, or long (Fig. 6E). For comparative analysis, the percentages of cells with no or short cilia and those with medium or long cilia were combined, to equal 100% of the epithelial cells. The percentage of HAECs with no or short cilia was higher in SA patients (Table 3, $p < 0.0001$), and inversely correlated with FEV1% predicted ($r = 0.64$, $p < 0.0001$) (Fig. 6F), suggesting that increases in short/no ciliated cells in asthmatic airways may worsen disease severity. 15LO1 protein similarly correlated with short/no cilia cell percentages (Fig. 6G, $r = 0.66$, $p < 0.0001$) and FEV1% predicted ($r = 0.51$, $p = 0.001$). Subsequent mediation analysis determined that 70% of the potential impact of 15LO1 on lung function (FEV1% predicted) was through an effect on ciliated cell phenotype (average causal mediation effect = 0.70, $p < 0.001$, Fig. 6H). Similarly, asthma symptoms as measured by Asthma Control Questionnaires (ACQ), positively correlated with 15LO1 ($r = 0.48$, $p = 0.006$) and the percentages of epithelial cells with no or short cilia ($r = 0.42$, $p = 0.02$). Gender did not influence any of these results. These data collectively suggest that 15LO1 pathway-mediated ferroptotic and mitophagic processes may contribute to a "damaged/altered" ciliated cell phenotype, which worsens asthma outcomes.

## Discussion

Mitochondria and their electron transport chains generate life-sustaining ATP to power cells for proliferation, differentiation and function. Excessive mitochondrial damage leads to cellular dysfunction, cell death and disease. Ferroptosis, with its Fe-dependent generation of phospholipid hydroperoxides, has long been associated with mitochondrial injury[1–3]. Yet the mechanisms for this damage or the cellular and functional responses, particularly in relation to human disease remain unclear. In this report, we identify a form of ferroptosis, driven by 15LO1-pathway derived oxidized phospholipid (PLox) species generated in mitochondrial fractions. While this ferroptosis initially leads to mitochondrial disruption, and fragmentation, the loss of $\Delta\Psi_m$ activates a PINK1/Parkin driven mitophagic process which is completed through OPTN activation of lipidated LC3. The convergence of these pathways protects cells from ferroptotic death and decreases release of mitochondrial DNA, a danger associated molecular pattern (DAMP) signal. These mechanistic processes are identified both in vitro in primary HAECs and in parallel ex vivo in fresh HAECs from asthmatic patients. Ultimately, these integrated ferroptotic-mitophagic processes impact cellular differentiation, alter ciliated cell phenotypes, and potentially influence asthma outcomes, with implications for other diseases, including cancer and chronic obstructive pulmonary disease.

Mitochondria are the major sources of intracellular ROS, primarily in the form of superoxide produced as by-products of oxidative phosphorylation and ATP generation. Pathologic levels of ROS can nonenzymatically oxidize phospholipids to generate hydroperoxyferroptotic lipids[48]. This process is typically reported to begin

*internally* following changes in $\Delta\Psi m$. Internal origins for ferroptosis are supported by studies in which direct activation of ferroptosis (RSL3 treatment, GPX4 inhibition, and others) leads to mitochondrial accumulation of PLox[49,50], and when mitochondria targeted anti-oxidants, like MitoTEMPO are reported to block ferroptosis[51]. Whether these ferroptotic changes can also occur in an "outside-in" manner has not been clear. Here, we report activation of a 15LO1-dependent pathway which programmatically targets mitochondria for ferroptotic membrane disruption and loss of $\Delta\Psi_m$. We identified 15LO1 in mitochondria fractions (along with the generation of ferroptotic PLox), but our data do not allow us to precisely determine mitochondrial location [inner or outer mitochondrial membranes (IMM/OMM)][52]. Both IMM and OMM contain high levels of PE[53], in association with free iron[54], but it is more likely enzymatically induced lipid peroxidation first occurs in the OMM. However, it is also possible that "nearby" 15LO1 in mitochondrial associated endoplasmic reticulum membranes[55] could similarly drive loss of $\Delta\Psi_m$, potentially through waves of ferroptosis[56,57]. These combined ferroptotic processes then induce mitochondrial swelling and eventual rupture as identified by TEM. Reversal of the mitochondrial effects by 15LO1 inhibition or KD confirms the critical role of this enzyme in this unique form of ferroptosis, as does the lack of effect of short term RSL3 on control cells with low 15LO1 levels. Combined, these studies identify enzymatically dependent compartmentalized mitochondrial damage in HAECs under high T2 conditions, as relevant to asthma. While the biologic purpose for this targeted mitochondrial disruption is as yet unclear, it could be relevant for physiological cellular differentiation processes, as has been reported for retinal pigmented epithelial cells[13] or as a self-protective process to target aging or damaged mitochondria for removal/recycling, or to prolong survival[58]. Our studies of HAECs under T2-Hi conditions support a central role for 15LO1 and accompanying mitophagy in the inhibition of ciliated cell differentiation.

To that end, this localized 15LO1 dependent ferroptotic mitochondrial disruption elegantly coordinates with lipidation of the quintessential autophagic membrane protein LC3 through shared PEBP1 activation pathways[11] to drive formation of mitophagic membranes and, in contrast to one previous study[59], limit cell death. Under IL-13/15LO1 high conditions, only activated LC3-II is present in mitochondrial fractions, indicating initiation of mitophagic membrane formation in mitochondrial compartments. Ferroptosis, as the driver for this mitophagy, is confirmed by lowering of LC3-II in mitochondrial compartments in response to 15LO1 KD. This coordinated ferroptosis with compensatory mitophagy has not been reported previously, suggesting that these combined effects are unique to 15LO1 mediated (outside-in) processes. Indeed, our results conflict with previous studies which suggest differing interactions between ferroptosis and mitophagy. Under non-enzymatic/cell stress conditions, mitophagy has been reported to trigger ferroptosis through ferritinophagy or even worsen cell death in response to ferroptosis[60,61]. The reasons for

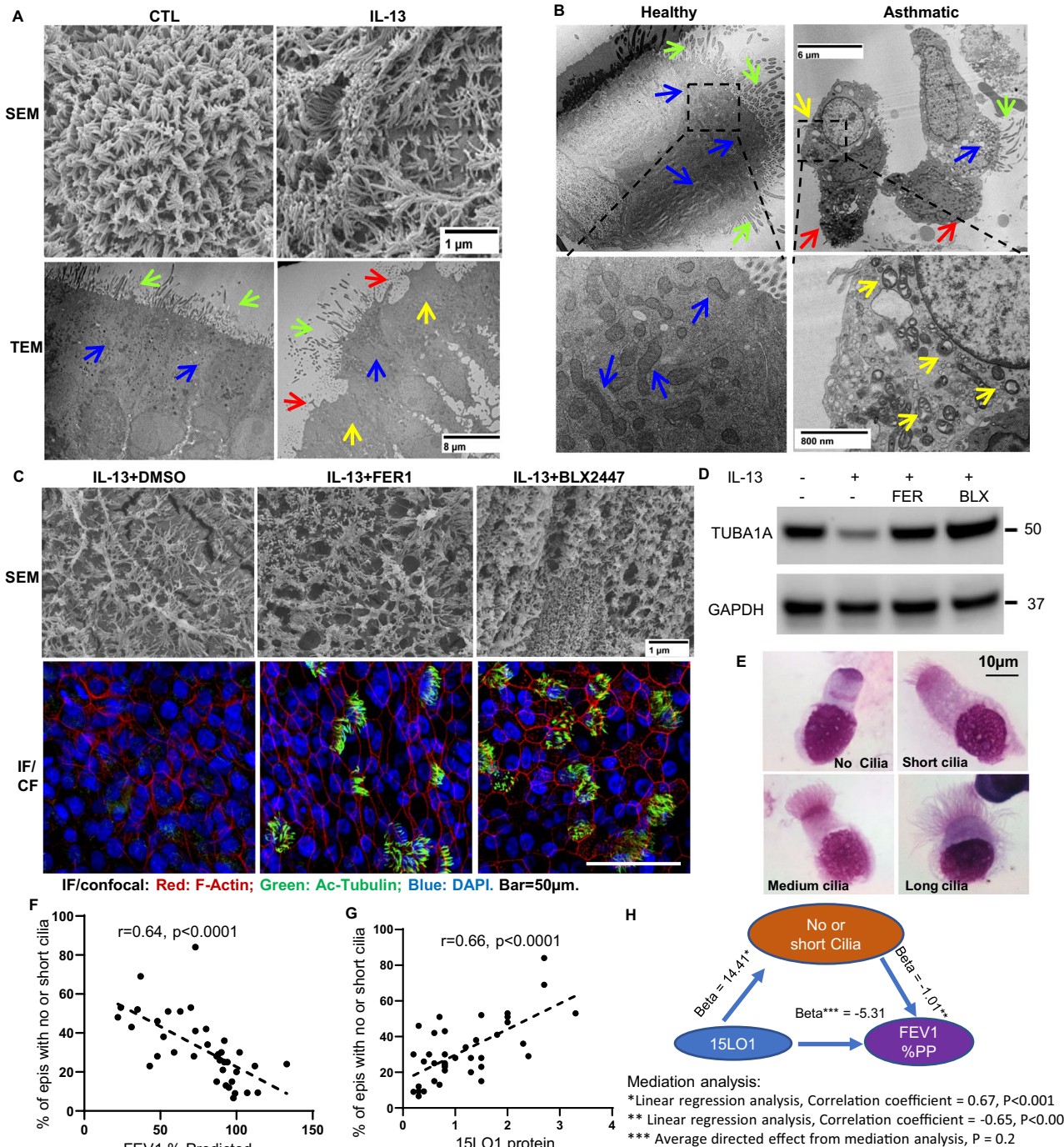

**Fig. 6 | Functional implication of 15LO1-dependent ferroptosis. A** Fewer ciliated cells under IL-13 compared to control conditions by SEM (upper panel) and TEM (lower panel) (n = 3 biological replicates). Green arrows: cilia; Red arrows: areas denuded of cilia. Blue arrows: "healthy" mitochondria; Yellow arrows: decreased mitochondria. **B** Loss of cilia and mitochondria, as well as presence of swollen mitochondria in fresh epithelial cells from a severe asthmatic compared to HC participant (TEM) (n = 3–5 samples from individual donors in each group). Green arrows: cilia; Red arrows: areas denuded of cilia. Blue arrows: "healthy" mitochondria; Yellow arrows: swollen mitochondria. **C** FER-1 (1 μM, 5 days) and/or BLX2477(2 μM, 5 days) prevent IL-13-induced cilia structural changes and ciliated cell loss by SEM (upper panel) and by tubulin staining (IF/CF, bottom panel).

**D** Under the same conditions, FER-1 and BLX2477 increase TUB1A expression by WB under IL-13 conditions. Representative images n = 3 biological replicates. **E** Representative images of fresh epithelial cells with varying cilia length (Diff-Quik staining) (n = 38 individual donors, see Table 3 for Demographics). **F** Percentages of total mature epithelial cells ex vivo with no/or short cilia negatively correlate with FEV1% predicted and (**G**) 15LO1 protein (by WB of same cells) positively correlate with percentage of cells with no/short cilia (Bivariate linear correlation).
**H** Mediation analysis supports that the majority of the potential impact of 15LO1 on lung function (FEV1% predicted) is through effects on ciliated cell phenotype. Data quantification and individual antibody images in Supplementary Fig. 6. Source data provided as Source Data file.

these differences are unknown, but could be related to differences between enzymatically or non-enzymatically initiated mitochondrial ferroptosis, or the 15LO1 dependent increases in de novo synthesis of LC3, which differ from canonical autophagy[11]. Finally, whether similar

interactions of ferroptosis with mitophagy occur with 15LO2 mediated processes and in other cell types remains unknown.

Formation of mitophagic membranes around damaged mitochondria requires several steps, beginning with identification and

**Table 3 | Demographics of Participants with Ciliated Cell Phenotype Data***

| | Healthy Controls | Mild-Moderate Asthma | Severe Asthma | Overall *p*-value |
|---|---|---|---|---|
| Age (mean±SD) | 40 ± 16 | 38 ± 15 | 44 ± 10 | 0.55 |
| Gender (W/M) | 7/0 | 12/4 | 12/3 | 0.36 |
| Self identi-fied race# | 1/5/1/0 | 3/13/0/0 | 3/10/1/1 | 0.72 |
| FEV1% pred (mean±SEM) | 97 ± 4 | 89±5 | 55 ± 6 | <0.0001 |
| FeNO (ppb) (mean±SD) | 13 ± 4 | 31±18 | 68 ± 76 | 0.036 |
| 15LO1 Protein | 0.7 ± 0.4 | 0.9±0.6 | 1.6 ± 0.9 | 0.007 |
| % No Cilia | 11 ± 4 | 15±8 | 30 ± 16 | 0.0006 |
| % Short Cilia | 5 ± 4 | 10±6 | 17 ± 6 | <0.0001 |
| % Short & No Cilia | 17 ± 7 | 24±10 | 47 ± 15 | <0.0001 |

ANOVA Analysis.
*18 participants overlap with Table 2, none with Table 1.
#Black/White/Mixed/Asian.
*SD* standard deviation, *SEM* Standard error of the mean, *IQR* Interquartile range.
W/M woman/man, self reported.
No biological usage in all participants.

targeting of damaged mitochondrial membranes. These activation steps can involve a number of receptor or non-receptor mediated process. Receptor mediated mitophagy involves up-regulation of receptors on OMM following loss of $\Delta\Psi m$[62]. In contrast, PINK1-Parkin induced mitophagy, perhaps the most commonly identified pathway, does not require intermediary receptors but rather is believed to be directly activated by loss of $\Delta\Psi m$.[46,47] Our studies support loss of $\Delta\Psi m$ under IL-13 conditions, potentially following lipid peroxidation of OMM by 15LO1-mediated processes[56,57]. The current data show 15LO1-dependent phosphorylation of Parkin under IL-13 conditions, supporting its phosphorylation in response to PINK1 expression in ferroptotically damaged OMMs and the specific link to mitophagy.

For PINK1-Parkin to interact with LC3 and initiate mitophagy, another adapter (bridge) protein is required. Both OPTN and NDP52 have been reported to provide that bridge[42], but in our study, OPTN was specifically bound to LC3 in Co-IP experiments. This specific binding of OPTN to LC3 further links the process with PINK1-Parkin. While other mitophagic pathways could still be involved, the broad impact of 15LO1 KD on pParkin, OPTN and LC3-II supports the primary engagement of PINK1-Parkin associated mitophagy in response to 15LO1 initiated compartmentalized ferroptotic mitochondrial membrane damage. Further, freshly brushed asthmatic epithelial cells exhibited similar coordinated upregulation in both OPTN and LC3, strongly supporting the activation of PINK1-Parkin-OPTN-LC3 pathway mediated mitophagy in asthmatic HAECs.

The airway epithelium in asthma is widely recognized to include over representation of goblet cells, and indeed, most epithelial studies in asthma have focused on goblet cell regulation and function. However, in addition to gain of goblet cells, there is also loss of ciliated cells, with T2 cytokines (and 15LO1) believed to play an important role in these changes[47,63-65]. Indeed, the differentiation of ciliated cells remains poorly understood, with most scRNAseq studies suggesting several different types of ciliated cells, both in vitro and ex vivo, with the trajectories for development only beginning to be understood[66-68]. Our human asthma data suggest that 15LO1-mediated mitochondrial ferroptosis could play a major role in the regulation and differentiation of airway epithelial cells. 15LO1 protein is highly expressed in both basal cells[30], progenitors for ciliated cells, and in ciliated cells. It is conceivable that expression/activity in basal cells induces mitochondrial loss limiting cilia development and function while promoting goblet cell differentiation. This concept is supported by the impact of

ferroptosis on early cell differentiation where basal cells predominate. Indeed, high percentages of columnar cells with abnormally short (or no) cilia in asthmatic airways were tightly linked to 15LO1 protein expression. Over-representation of goblet cells (and mucus) may push the remaining ciliated cells to work (beat) even more rigorously, but also leading to excessive mitochondrial stress and more susceptibility to further damage. Interestingly, IL-13 has also been reported to enhance mucus secretion from goblet cells in an autophagic dependent process[69], which could further increase ciliated cell stress. Removal and subsequent recycling of these stressed and/or damaged organelles through combined ferroptosis and mitophagy could help preserve some level of ciliated cell function, but at the expense of increasing ROS generation. However, this process could also be occurring under normal physiologic conditions. Indeed, we observed LC3 colocalized with OPTN in some fresh ciliated cells from healthy participants, suggesting that mitophagy (in response to cell aging/senescence) may be a "healthy" process to extend the life of normal ciliated cells.

Importantly, under 15LO1 Hi conditions this pro-survival mitophagy may be compromised by the continuous generation of ROS/PLox, with levels well beyond the capacity of self-preservation. These combined ferroptotic-mitophagic attempts to preserve the ciliated cells with compromised ciliary function could explain the overall loss of cilia under high 15LO1 conditions and the link to worsened asthma outcomes. In fact, prolonged preservation of damaged cells could lead to failure of their clearance through apoptosis or other death pathways[70,71]. Ongoing ferroptosis and mitophagic responses could further perpetuate loss of intracellular anti-oxidants, like GSH, which, as we reported, would also enhance secretion of Type 2 signature proteins[10]. The known expression of 15LO1 in basal cells could also potentially limit or alter their differentiation into ciliated cells which require abundant functional mitochondria, and which could also contribute to the lower percentages of ciliated cells in asthmatic airways[8,10,30]. Inhibiting mitophagy under these conditions would enhance cell death and potentially promote generation and differentiation of more robust and normal epithelial cells.

Although mitophagy has been previously reported in severe asthmatic fibroblasts[72], and in primary human bronchial epithelial cells[73], its functional implications in asthma remain controversial[74,75]. Depending on the type of selective autophagy, it can promote cell death, often through indirect mechanisms which favor biochemical pathways that facilitate pro-ferroptotic oxidative reactions[74,75], or, as we suggest here, act as a cytoprotective mechanism which selectively removes damaged or dysfunctional cellular components. Our results suggest that in the face of ongoing ferroptosis, mitophagy preserves mitochondrial components which may serve to stabilize ciliated cells and their function. Autophagy has also been previously linked to cilia dysfunction in chronic obstructive pulmonary disease, (COPD), particularly in relation to smoke induced oxidative stress, but the relation to human disease is less clear[76]. PINK1-Parkin was also reported to be activated in respiratory epithelium in chronic obstructive pulmonary disease (COPD)[37-40], similarly in response to cigarette smoke and its toxicants. Interestingly, a subset of COPD patients is known to have increases in Type-2 immune pathways and eosinophilia, with a recent study indicating efficacy of the IL-4Ra antibody dupilumab in COPD[77]. Thus, it is conceivable that a compartmentalized ferroptosis may be driving PINK1-Parkin mitophagy in COPD, as well. Whether similar changes are occurring in other epithelial compartments (i.e., gastrointestinal) or even in cancer, is unknown but further studies are needed.

In conclusion, the work reported here identifies a convergence of two fundamental cell death and survival pathways with implications for epithelial cell differentiation, function, and human disease outcomes. Further understanding of compartmentalized ferroptosis and its mitophagic response could lead to approaches to normalizing diseased epithelium, of relevance to asthma and potentially to other diseases.

## Methods

### General statement

This research complies with all relevant ethical regulations and study protocol that approved by University of Pittsburgh Institutional Review Board (IRB). All studies in which quantifiable imaging was performed were done in a blinded or masked manner by the readers.

### Reagents and antibodies

Antibodies against LC3 (rabbit IgG) was purchased from Sigma-Aldrich (St. Louis, MO). 15LO1 (rabbit IgG) was from Abnova (Walnut, CA). PINK1 (rabbit IgG), LC3B (rabbit IgG) and GAPDH (goat IgG) was from NOVUS (Littleton, CO). pParkin (rabbit IgG) and OPTN (rabbit IgG) were from Proteintech (Rosemont, IL) PEBP1 (mouse IgG) and OPTN (mouse IgG) was from Santa Cruz (Santa Cruz, CA). GPX4 (rabbit IgG), MTCO2 (mouse IgG) and Total OXHPOS (mouse IgG) were from Abcam (Cambridge, MA). ATP synthase (Mouse IgG) was from Invtrogen (Carlsbad, CA). The siALOX15 DsiRNA™ was purchased from IDT (5′-UGUUUUACGCUAAAGAUGGAAAAGA-3′; 3′-CAACAAAAUGCGAUUUC UACCUUUUCU-5′, Coralville, IA), and *Lipofectamine* transfection reagent from Thermo Fisher (Rodkford, IL). Basic Epithelial Growth Medium (BEGM) cell culture medium and supplements were purchased from Lonza (Basel, Switzerland). Recombinant human IL-13 was purchased from R&D Systems (Minneapolis, MN). BLX2477, a highly specific inhibitor of 15LO1, was a kind gift from Dr. Hans-Erik Claesson[78].

### Sources of HAEC for in vitro and ex vivo studies

HAECs were obtained by bronchoscopic brushing of asthmatic and healthy control (HC) airways as previously described[79]. All participants were recruited as part of the Immune-epithelial Cell Interactions in Severe Asthma (P01 AI106684 and P01AI106684-06A1)[80]. All asthmatic participants met American Thoracic Society (ATS) criteria for asthma and included mild to severe asthmatic patients, while HCs were without respiratory disease and had normal lung function[81,82]. No participant smoked within the last year or > 5 pack years or was studied within 4 weeks of an asthma exacerbation. All participants were extensively evaluated through lung function testing (race-neutral FEV1 equations), FeNO, allergy testing and questionnaires. Exacerbation history was determined by questionnaires. Severe asthma was defined by ERS/ATS criteria including use of high dose inhaled and/or systemic corticosteroids, in combination with a 2nd controller to maintain control or who remained uncontrolled[81]. Mild-moderate asthma (Mild/Mod) participants were all those who did not meet severe asthma criteria. Self-reported gender (man/woman) was used in this study. The study was approved by the University of Pittsburgh Institutional Review Board and all participants gave informed consent. The characteristics of these 60 (total) participants are included in three separate demographics tables depending on study (Tables 1-3).

### Primary human airway epithelial cell culture in air–liquid interface, DsiRNAtransfection

HAECs were cultured in air–liquid interface (ALI) under serum-free condition as previously described[31,80]. Briefly, fresh bronchoscopic brushing primary HAECs or banked samples were cultured under immersed condition for proliferation. When 80-90% confluent, cells are trypsinized and plated at $5 \times 10^4$ cells per well on 12-well Transwell plate for submerged stage culture by adding 200 µl culture mdium to upper insert and 1000 µl culture medium to lower chamber [BEBM/DMEM at 50:50, supplemental with 4 g/ml Insulin, 5 pg/ml Transferrin, 0.5 µg/ml, Hydrocortisone, 0.5 µg/ml Epinephrine, 52 µg/ml Bovine hypothalamus extract, 50 µg/ml, Gentamicin, 50 ng/ml Amphotericin, 0.5 µg/ml albumine bovine, 80 nM ethanolamine, 0.3 mM, $MgCl_2$, 0.4 mM $MgSO_4$, 1 mM $CaCl_2$, 30 ng/ml retinoic acid and 0.5 ng/ml Epithelial Growth Factor (EGF)]. When cells reached 100% confluence,

cells went into ALI culture for by reducing the upper volume to 50 µl with the lower volume remaining at 1.0 mL full medium. For TOM20 staining, HAECs were stimulated with IL-13 (10 ng/ml) for 7 days from the beginning of ALI. For ATPsynthase staining, HAECs were cultured under ALI for 7 days and then stimulated with IL-13 for 7 days. For RSL3 treatment, RSL3 (10 µM) was added for 2 h, with DMSO as control. For TEM analysis of RSL3 impact, HAECs were cultured with media alone for 9 days and then stimulated with IL-13 for 5 days, with/without addition of FER-1 (1 µM, for 5 days) and/or BLX2477(2 µM for 5 days). RSL3 (10 µM) was then added 2 h prior to harvest for TEM. DsiRNA transfection was performed using Lipofectamine transfection reagent. Briefly, 50 nM DsiRNA was pre-mixed with 3 µl/well Lipofectamine transfection reagent for 20 min at room temperature before pooled together with HAECs suspension and seeded onto transwells for incubations. After 24 h, the transfection mixture was removed, and cells were switched to ALI culture for 7 days. Cells were stimulated with IL-13 (10 ng/ml) under ALI culture for 7 days.

### Total DNA Isolation and mitochondrial DNA Quantification

Total DNA was extracted from cell lysate using the DNeasy Blood &Tissue Kit (Qiagen, Germantown, MD). Quantification of mitochondrial DNA was assayed by real-time qPCR for the human MT1 ND1 gene using Taqman gene expression assays (Fisher Scientific, USA). The number of MT ND1 was determined based on a standard curve developed from serial dilutions of a commercially available DNA plasmid and the mitochondria DNA concentration was expressed as ng/ug per protein as previously described in ref. 11.

### Co-immunoprecipition and Western blot

Total proteins were harvested in protein lysis buffer (50 mM Tris-HCl, 150 mM NaCl, 10 mM EDTA, 1.0 % NP-40). After denatured at 95°C in 5XSDS sample buffer for 5 min, the samples were run in 4%–12% sodium dodecyl sulphate–polyacrylamide (SDS-PAGE) gels (Invitrogen) and transferred onto polyvinylidene difluoride membrane (Invitrogen). After the membrane was blocked with 5% skim milk, target proteins were immunodetected using primary antibodies Membranes were developed using an Amersham Imager 600 (GE Healthcare Life Sciences) with SuperSignal West Femto Maximum Sensitivity Substrate (Thermo Fisher Scientific, 34096). Densitometry analysis was performed using Amersham Imager 600 and ImageJ software (NIH).

For co-immunoprecipitation, cells were washed with cold PBS then lysed in mild protein lysis buffer (50 mM Tris-HCl, 150 mM NaCl, 10 mM EDTA, 0.2 % NP-40) with protease inhibitors. The cell lysate was pre-cleaned with protein A agarose beads and incubated with pull-down primary antibody overnight. Protein A agarose beads were added and incubated for 1 h at 4 °C. The immunoprecipitates were washed and boiled in 2X SDS sample buffer for 5 min. After centrifugation, the supernatant protein was separated on 12% SDS-PAGE gels for Western Blot using primary and secondary antibodies generated from species different to the pull-down primary antibody.

### Mitochondrial membrane potential analysis

Cells were suspended in Basic epithelial Growth Medium (BEGM) with 100 nM TMRM and incubated in a CO2 incubator for 30 min. Cells were then washed one time with medium, resuspended in 500 µL of medium and analyzed using NovoExpress flow cytometer (Agilent) with excitation at 488 nm and an emission at $572 \pm 28$ nm (PE equivalent bandpass filter).

### Immunofluorescence confocal microscopy and object based co-localization analyses

Cells cultured on ALI trans-well membrane were directly fixed in 2% paraformaldehyde at 4 °C for 15 min, and permeabilized with 0.1%

Triton X-100 in PBS for 15 min before incubation with the primary antibodies of interest. The cells were washed again and incubated with AlexaFlour 555 and 488 secondary antibodies. 4'6-diamidino-2-phenylindole (DAPI) nuclear stain and no-primary antibody controls were done at the same time. For ex vivo studies, cytospins of freshly brushed primary epithelial cells were fixed in 2% paraformaldehyde at 4 °C for 15 min, permeabilized and rehydrated before incubation with primary antibodies for IF staining. The cells were washed, mounted and confocal z-stacks (150 nm optical sections) collected using a Nikon A1 Confocal equipped with a 60X, 1.4NA objective. The confocal z-stacks were processed using blind deconvolution (10 iterations, NIS Elements, Nikon Inc., Melville NY) followed by object-based co-localization analysis. Puncta (objects) were segmented using 3D spot detection (based on intensity, size and shape) followed by co-localization analysis using bolean "having" statements. For measurements of mitochondrial fragmentation and abundance, the confocal datasets were deconvolved using Nikon Elements (Nikon Inc., Melville, NY) and then imported into Imaris (Bitplane Zurich, Switzerland) for surface rendering and calculation of mitochondrial volumes. Mitochondrial fragmentation was assessed using the sphericity parameter (defined as the ratio of the surface area of the given object to the surface area of a sphere with the same volume as the given object).

## Transmission electron microscopy

Freshly brushed human airway epithelial cells (HAECs, ex vivo) in PBS were centrifuged at 600 rpm for 10 min, and the pellets were fixed in 2.5% glutaraldehyde in 0.01 M PBS (8 gm/l NaCl, 0.2 gm/l KCl, 1.15 gm/l $Na_2HPO_4 \cdot 7H_2O$, 0.2 gm/l $KH_2PO_4$, pH 7.4) for 1 h at 4 °C. Pellets were then washed in PBS three times then post-fixed in aqueous 1% osmium tetroxide, 1% $Fe_6CN_3$ for 1 hr. Cells grown on tissue culture Transwells ALI were directly fixed on the membrane insert and then cutting off for subsequent process. Samples were then washed 3 times in PBS then dehydrated through a 30–100% ethanol series then several changes of Polybed 812 embedding resin (Polysciences, Warrington, PA). Blocks were cured overnight at 37 °C, then cured for two days at 65 °C. Ultrathin sections (60 nm) of the pellets were obtained on a Riechart Ultracut E microtome, post-stained in 4% uranyl acetate for 10 min and 1% lead citrate for 7 min. Sections were viewed on a JEOL JEM 1400 FLASH transmission electron microscope (JEOL, Peobody MA) at 80 KV. Images were taken using a bottom mount AMT digital camera (Advanced Microscopy Techniques, Danvers, MA).

## Scanning electron microscopy

Cells grown on tissue culture Transwells were washed with PBS 2x for 5 min, then fixed in 2.5% glutaraldehyde in 0.01 M PBS (8 gm/l NaCl, 0.2 gm/l KCl, 1.15 gm/l $Na_2HPO_4 \cdot 7H_2O$, 0.2 gm/l $KH_2PO_4$, pH 7.4) for 1 hr at 4 °C. Inserts were then washed in PBS three times then post-fixed in aqueous 1% osmium tetroxide for 1 hr. Inserts were washed 3 times in PBS then dehydrated through a 30-100% ethanol series and further dehydrated by three additional 15 min washes with absolute ethanol. Next, the samples were washed in Heximethyldisilizane (HMDS) for 15 min and then removed to air dry. Samples were cut out from the supporting wells and then mounted onto aluminum stubs, grounded with silver paint, and sputter coated with 3.5 nm gold/paladium (Cressington Sputter Coater Auto 108, Cressington, Watford, UK). Samples were viewed in a JEOL JSM-6335F scanning electron microscope (Peabody, MA) at 3 kV.

## LC-MS/MS analysis of oxygenated phospholipids

Lipids were extracted using the Folch procedure, and phosphorus was determined by a micromethod[5,83]. As CL and PE are specifically related to mitochondrial dysfunction and ferroptosis, respectively, their polyunsaturated species as well as oxygenated metabolites (hydroxy-

and hydroperoxy-derivatives) were analyzed by LC/MS. A Dionex Ultimate 3000 HPLC system (normal phase column, Luna 3 μm silica C18(2) 100 Å, 150 × 2.0 mm, Phenomenex) coupled on-line to an Orbitrap Fusion Lumos (ThermoFisher Scientific) mass spectrometer was employed. The column was maintained at 35 °C. The analysis was performed using gradient solvents (A and B) containing 10 mM ammonium formate at a flow rate of 0.2 mL/min. Solvent A contained isopropanol/hexane/water (285:215:5, v/v/v), and solvent B contained isopropanol/hexane/water (285:215:40, v/v/v). All solvents were LC/MS grade. The column was eluted for 0–23 min with a linear gradient from 10% to 32% B; 23 to 32 min with a linear gradient of 32% to 65% B; 32–35 min with a linear gradient of 65% to 100% B; 35–62 min held at 100% B; 62–64 min with a linear gradient from 100% to 10% B; followed by an equilibration from 64 to 80 min at 10% B. Analysis was performed in negative ion mode at a resolution of 140,000 for the full MS scan in a data-dependent mode. The scan range for MS analysis was m/z 400 to 1800 with a maximum injection time of 128 ms using 1 microscan. An isolation window of 1.0 Da was set for the MS scans. Capillary spray voltage was set at 3.5 kV, and capillary temperature was 320 °C. The S-lens RF level was set to 60. Ion source conditions were set as follows: spray voltage = 4 kV, sheath gas = 20 (arbitrary unit), auxiliary gas = 4 (arbitrary unit), sweep gas = 0 (arbitrary units), transfer tube temperature = 300 °C, RF-lens level = 50%. Analysis of raw LC/MS data was performed using software package Compound Discoverer 2.0 (Thermo Fisher Scientific) with an in-house-generated analysis workflow and oxidized-phospholipid database. Briefly, peaks with a signal to noise (S/N) ratio of greater than 3 were identified and searched against the oxidized-phospholipid database. Lipids were further filtered by retention time and confirmed by a fragmentation mass spectrum. 15 HpETE-PE and 15 HETE-PE were further characterized in mitochondrial fractions using MS/MS (Supplementary Fig. 7A, B). A mixture of deuterated phospholipids consisting of 1-hexadecanoyl(d31)-2-(9Z-octadecenoyl)-sn-glycero-3-phospho-ethanolamine (PE(16:0D31/18:1)), 1-hexadecanoyl(d31)-2-(9Z-octadecenoyl)-sn-glycero-3-phosphocholine (PC(16:0D31/18:1)), 1-hexadecanoyl(d31)-2-(9Z-octadecenoyl)-sn-glycero-3-phospho-serine (PS(16 :0D31/18:1)), 1-hexa-decanoyl(d31)-2-(9Z-octadecenoyl)-sn-glycero-3-phosphate (PA(16:0D31/18:1)), 1-hexadecanoyl(d31)-2-(9Z-octadecenoyl)-sn-glycero-3-phosphoglycerol (PG(16:0D31/ 18:1)), 1-hexadecanoyl(d31)-2-(9Z-octadecenoyl)-sn-glycero-3-phospho-(1'-myo-inositol) (PI(16:0D31/18:1)) (Avanti Polar Lipids) was used as internal standards. 1,1',2,2'-tetramyristoyl-cardiolipin (sodium salt) (Avanti Polar Lipids) was used as cardiolipin internal standard. Values for m/z were matched within 5 ppm to identify the lipid species.

## Isolation of mitochondrial fractions

Mitochondrial fractions were isolated using the Mitochondria Isolation Kit for Cultured Cells (ab110171, Abcam) with some modifications of the manufacturer's instructions. Briefly, HAECs were collected from transwell membranes with a cell lifter and pelleted by centrifugation at 700 g for 5 min. The cells were frozen, thawed to weaken cell membranes, resuspended in Reagent A and then incubated on ice for 10 min. Cells were transferred into a pre-cooled Dounce Homogenizer for 3–5 strokes. Homogenates were centrifuged at 1000 g for 10 min at 4 °C and saved as supernatant #1. The pellet was resuspended in Reagent B and cell rupturing repeated. These homogenates were centrifuged and saved as supernatant #2. Supernatants #1 and #2 were combined and centrifuged at 6000 g for 15 min. The pellet was washed x2 with PBS to remove contamination and centrifuged at 6000 g for 15 min. The pellet was resuspended into Reagent C supplemented with Protease Inhibitors and the aliquots frozen at −80 °C until use.

The mitochondrial and cytosolic fractions were evaluated for relative purity by measuring both compartmental lipids using LC-MS/MS and proteins (Western blot) of interest. Cardiolipins (CLs) are

uniquely characteristic of mitochondria[17], while phosphatidylserines (PSs) are absent[84]. Additionally bis(monoacylglycero)phosphatidic acid species (BMPs) are specific to lysosomes[85]. Thus, the CL/PS ratio is a sensitive and specific marker of mitochondrial purity. Supplementary Fig. 8A demonstrates highly significant differences between the CL/PS ratios in the mitochondrial and cytosolic fractions. The CL/BMP ratios further confirm limited contamination of mitochondrial fractions by cytosolic lysosomes (Supplementary Fig. 8B). To exclude potential contamination with peroxisomes in the mitochondrial fraction, we assessed the presence of plasmalogens[86], such that the cytosolic compartment would be enriched in plasmalogenic forms of PE whereas mitochondrial PE is enriched with di-acylated-molecular species of PEs[87]. Indeed, the cytosolic fraction was enriched in PEp as evidenced by the highly significant differences in PEp/PEd ratios in the cytosol vs mitochondria, respectively (Supplementary Fig. 8C). These data collectively support the relative purity of the mitochondria fractions.

## Quantification of proteins in cell compartments

Normalization for quantification by WB is often challenging. As IL-13 stimulation decreases intracellular mitochondria numbers/volume, indexing to a mitochondrial marker, as a housekeeping gene, is not appropriate. Accordingly, to evaluate quantitative changes in mitochondrial proteins under IL-13 conditions, the entire mitochondria fraction obtained after separation was loaded onto the gels, each of which was derived from the original total cell protein (as defined by GAPDH levels). The proteins in the total mitochondrial fraction were then indexed to the starting total GAPDH in the cell (as measured in the cytosolic fraction), and as has been previously reported and performed[88–92].

## Epithelial cilia staining and characterization

Cytospins from freshly brushed HAECs of healthy and asthmatic participants were stained with Diff-Quik, and 100 mature epithelial cells blindly counted and characterized for the presence/type of cilia. Cilia length was defined as short, medium, or long. The data were presented as the percentage of total cell counted.

## LDH assays

Cell-free culture supernatants and cell lysates were collected respectively measured for LDH using the Lactate Dehydrogenase Assay Kit (Abnova, Walnut, CA). The results are presented as percent (%) released determined as the following: % released = n(LDH in supernatants)/[n(LDH in supernatants)+ n(LDH in cell lysis)] X100%. $n$ = LDH activity unit.

## Statistical analysis

Statistical analysis was performed using J GraphPad Prism software version 7. Data that were normally distributed were represented as means ± SEM. Difference between conditions and subject groups were analyzed for distribution and then compared using one way ANOVA, paired or unpaired t-testing. FeNO was not normally distributed and comparisons were made with non-parametric (Spearman's Rho) testing. Cells from specific donors under 2 conditions were compared using matched-paired analysis. Ex vivo studies were controlled for age and gender, The "n" is insufficient to control for gender in vitro studies. Pearson's correlation was employed to analyze relationships among the continuous data. Two-sided test were used in all comparison and $p$-values of < 0.05 were considered statistically significant.

## Reporting summary

Further information on research design is available in the Nature Portfolio Reporting Summary linked to this article.

## Data availability

Source data are provided within this paper. Raw flow cytometry data can be accessed from the Elsevier's Mendeley Data Repository (DOI:10.17632/5jtbgzn5pv.1). MS data and any other study related data not presented in this manuscript are available upon request. Source data are provided with this paper.

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

## Acknowledgements

Dr. Seyed Mehdi Nouraie for statistical support, Taylor Nee, MS for her help with bronchoscopy studies and clinical data. NIH R01 AI145406 (SEW), P01AI106684-06A1 (SEW), R01 HL153058 (SEW). KY received grant support from the Japanese Society of Allergology.

## Author contributions

Kazuhiro Yamada, MD, PhD Conceptualized and performed most of the experiments, analyzed and interpreted data, developed figures and text for the manuscript. Claudette St. Croix, PhD Developed methodologies, performed many of the Imaging studies and quantified the results. Contributed to figure and manuscript development. Donna B Stolz, PhD Developed methodologies and analyzed electron microscopy studies. Contributed to figure and manuscript development. Yulia Tyurina, PhD Developed methodologies and analyzed mass spectroscopy data. Contributed to figure and manuscript development. Vladimir Tyurin, PhD Developed methodologies and performed/analyzed the mass spectroscopy data. Contributed to figure and manuscript development. Laura B. Bradley BS Performed studies of freshly brushed epithelial cells and analyzed the data and processed the fresh and cultured cells. Alexander A. Kapralov, PhD Performed and analyzed the mitochondrial membrane potential experiments. Developed the accompanying figures and text, as well as general manuscript development. Yanhan Deng, MD, PhD Developed methodologies, fractionated cells and analyzed the mitochondria DNA. XIuxia Zhou, PhD Performed experiments. Qi Wei, MPH Performed experiments. Contributed to figure development. Bo Liao, MD, PhD Performed experiments, and contributed to figure development. Nobuhiko Fukuda, MD, PhD. Performed experiments and contributed to figure development. Mara Sullivan BS. Performed electron microscopy studies and contributed to figure development. John Trudeau, BA. Processed fresh and cultured epithelial cells for all the studies. Anuradha Ray, PhD Provided scientific and financial support for the study, reviewed and edited the manuscript. Valerian Kagan, PhD Provided intellectual and scientific input. Oversaw all mass spectroscopy work. Contributed to manuscript and figure development and final editing. Jinming Zhao, PhD. Provided the intellectual and scientific basis for the study. Oversaw all the cell culture studies. Performed experiments, analyzed data and developed and edited the main figures and text. Sally E. Wenzel, MD. Provided intellectual and scientific input which contributed to the overall hypotheses and to the final manuscript. Oversaw all aspects of the study. Contributed to manuscript development, editing and finalization.

## Competing interests

Dr. Wenzel is principal investigator on an investigator initiated/single center study of mucociliary clearance and the IL-4R antibody dupilumab in asthma patients funded by Regeneron. No other authors declare any conflicts.

## Additional information

[1]Department of Environmental and Occupational Health, School of Public Health, University of Pittsburgh, Pittsburgh, PA 15261, USA. [2]Department of Respiratory Medicine, Graduate School of Medicine, Osaka Metropolitan University, Osaka 545-8585, Japan. [3]Department of Cell Biology, University of Pittsburgh, Pittsburgh, PA 15261, USA. [4]Department of Rheumatology and Immunology, Tongji Hospital, Huazhong University of Science and Technology, Wuhan 430030, China. [5]Department of Otolaryngology-Head & Neck Surgery, Tongji Hospital, Huazhong University of Science and Technology, Wuhan 430030, China. [6]Department of Pulmonology, Yokohama City University Graduate School of Medicine, Yokohama 236-0004, Japan. [7]Department of Medicine, School of Medicine, University of Pittsburgh, Pittsburgh, PA 15260, USA. ✉e-mail: JinmingZhao@pitt.edu; swenzel@pitt.edu

