## [Peer Review File · Nature Communications]

Compartmentalized Mitochondrial Ferroptosis Converges with Optineurin-mediated Mitophagy to Impact Airway Epithelial Cell Phenotypes and Asthma OutcomesREVIEWER COMMENTS

Reviewer #1 (Remarks to the Author):

The manuscript by Yamada et al., describes the induction of ferroptosis via IL-13-induced 15LO1 production that leads to increased mitophagy. The confocal microscopy images are interesting to provide visualization of the data. Providing some human data from asthmatics provides some translation of the in vitro work in HAECs.

Main Points:

1. Ferroptosis has been shown to have a role in severe asthma. Other than showing that IL-13 decreases GPX4, determination of ferroptosis was not measured - the availability of redox-active iron and the oxidation of PUFA-containing phospholipids. Is the degree of ferroptosis different in mild vs severe asthma? Does apoptosis actually occur?
2. Mitophagy has been shown to have a role in severe asthma. An increase in PINK, Parkin, and LC3-II do not necessarily indicate an increase in mitophagy. Inhibition of mitophagy could show similar results because these markers will accumulate when they are not properly degraded. Need to measure autophagy activity (autophagy flux).
3. Mitophagy is induced by mitochondrial dysfunction. The two major measures of dysfunction are increased mtROS and a reduction/loss in membrane potential. Does IL-13 or 15LO1 mediate these abnormalities?
4. Most of the Western blots and confocal images are not quantified, so the reproducibility of the data is questionable.
5. Some controls are missing in several figures, such as Fig. 1G, Fig. 2C, Fig. 3C, 3F, & 3H, 4F, 4G, 4I, & 4J.
6. How does 15LO1 get into mitochondria, and what mitochondrial compartment - OMM, IMS, IMM, or matrix - does it locate?
7. Because IL-13 appears to only effect a few cells in the monolayer - Fig. 2B, Fig. 3E & 3H - is the phenotype of the entire monolayer changed?
8. How does ferroptosis, mitophagy, or 15LO1 change cilia number, length, or function?
9. Many of the conclusions in the figures are overstated or incorrect. For example, LC3-II is actually increased in Fig. 3G with siALOX, and COXII is present in the mitochondrial fraction with or without IL-13 in Fig. 2A.
10. Many of the Western blots are poor quality. There is a halo around the band.

Reviewer #2 (Remarks to the Author):

This study by Yamada et. al sought to define the mechanism by which 15 lipoxygenase-1 15LO1 (or ALOX15) which is highly expressed in the airway epithelium of individuals with asthma intersects with ferroptosis, focusing on mitochondria. The authors first show that IL-13 treatment of HAECs resulted in a reduction in the number and length of mitochondria. Upon an acute dose of the ferroptosis inducer RSL3, mitochondrial fragmentation was enhanced in IL-13-treated HAECs and prevented by dicer siRNA to ALOX15. Using mitochondrial enriched fractions, the authors then show that 15LO1 expression increases in both mitochondrial enriched and cytosolic fractions upon 1L-13 stimulation aligned with GPX4 expression. Immunofluorescence (IF) also showed some localization between 15LO1 and ATPsynthase. These IF findings were recapitulated in HAEC from individuals with asthma. Next HPLC/MS was used to measure oxidized lipids in mitochondrial enriched fractions upon IL-13 stimulation showing increased levels of 15OL1 oxidized PE species. Total oxidized PE's were also higher in whole cell lysates from HAECs from individuals with asthma. IL-13 stimulated autophagy, defined as increased LC3-II followed the induction of 15LO1 protein, and co-localization of these two proteins was observed by IF as well as in mitochondrial enriched fractions. Finally, the authors show that PINK1-Parkin as well as OPTN are activated in HAECs upon 1L-13 stimulation and that mitophagy membranes were observed in ciliated epithelial cells by TEM. Mitophagy markers were also higher in freshly brushed HAECs from asthmatics with enrichment in ciliated epithelial cells and this correlated with those cells that had shorter cilia- an effect that correlated with more severe asthmatic disease.

This work builds on prior published observations by this group that human airway epithelial cells (HAECs) release extracellular mtDNA release upon stimulation with inducers of ferroptosis in the presence of IL-13 and that the autophagy regulator LC3-II limits such an increase. This study is important in delineating the role of mitochondrial biology in airway epithelial cells in asthma pathogenesis. Strengths of this study include the use of primary human AECs grown at ALI, as well as fresh AECs from healthy controls and individuals with asthma with some clinical characterization. The use of subcellular HPLC/MS measurements in crude mitochondrial fractions is innovative. There is strong translational relevance, and the use of IL-13 ALI model appears robust. The link between cilia length/function and 15LO1 is novel and of interest. Limitations to this study include the use of the method used to generate "mitochondrial" fractions, the lack of clarity surrounding the role of ferroptosis in IL-13-induced autophagy/mitophagy, some conceptual overlap with prior publications regarding the role of autophagy/mitophagy in controlling cilia length and the role of autophagy in asthma pathogenesis. There are three major proposed mechanisms (ferroptosis via 15LO1, autophagy/mitophagy, and cilia shortening/loss) with neither one explored robustly in detail. To expand on these concerns.

Major concerns.

1. The Abcam mitochondrial fractionation kit is a useful tool; however, the mitochondrial fraction produced may not be as pure as expected with peroxisome and lysosomes present (Curr Protoc Cell Biol 2001 May; Chapter 3:Unit 3.4) and is usually termed the “crude” mitochondrial fraction. Given the important role of lipids. (and possible ferroptosis) in both of these organelles, it is imperative that the authors prove there is no contamination in their fractions using organelle markers specific for peroxisomal proteins or lysosomal proteins or use an alternative method such as gradient density centrifugation or the generation of mitoplasts (if interested in the matrix). Given the reliance of the majority of findings on this method, it is hard to deduce how mitochondrial-specific these findings are.
2. The use of the low dose acute treatment of RSL3 in Figure 1 is used to claim that compartmentalized ferroptosis may occur with loss of ALOX15 in HAECs reducing the effect of IL-13/RSL3 on mitochondrial numbers structure and autophagy. Yet for the rest of the manuscript, RSL3 is not used and other classic inducers of ferroptosis or inhibiting ferroptosis by classical means are not linked directly to il-13-induced mitophagy/autophagy. Can ferroptosis inhibitors (or lowering iron) protect HAECs from il-13-induced mitophagy, cilia shortening/loss etc.
3. Lack of readouts of mitochondrial function such as membrane potential, respiration, movement etc.
4. Which cell type is important? It is clear that the ciliated epithelial cell is the focus of this study, but what proportion of HAEC cultures at ALI are ciliated, goblet and basal remains elusive- i.e., the key experiments that demonstrate co-localisation of 15LO1 in mitochondrial fractions – is this in ciliated cells only? Or does this occur in goblet and basal cells too? The role for mitochondria in each of these diverse cell types is very different; so, is this proposed pathway also specific to one cell type or all?
5. Conceptually the link between IL-13 and autophagy has been described (Thorax. 2020 Sep;75(9):717-724) with autophagy essential for airway secretion of goblet cells in response to il-13 (Autophagy. 2016 Feb; 12(2): 397–409.). Cilia length and function in the airway epithelium has also been linked to autophagy (J. Clin. Invest. 123, 5212–5230 10.1172/JCI69636), and increased autophagy-related 5 gene expression is associated with collagen expression in the airways of refractory asthmatics (Front. Immunol. 8, 355 10.3389/fimmu.2017.00355).

Minor concerns.

1. The immunoblots presented using these mitochondrial enriched fractions should also include a mitochondrial housekeeping protein in addition to proteins like ETC proteins. i.e. Figure 2A- need tim23 or tom 20 loading control
2. The authors mention in Figure 1A and Figure S1A and B that mitochondria volume and size under both 108 short (identified by TOM20) and longer (identified by ATPsynthase) term cultures are used- it is unclear why use two separate markers – clarification would be helpful here.

3. Figure 1E and F- no RSL3 only controls
4. mtDNA measurements not normalized to total nuclear DNA – so unclear of loss of mtDNA due to loss in cells.
5. Lack of quantification of TEM images.
6. Scale bars lacking on all images.
7. It is unclear how many biological and/or technical replicates are used in all panels of the figures.

Reviewer #3 (Remarks to the Author):

Yamada et al investigated A 15-Lipoxygenase-1 dependent pathway drives a mitochondrially-targeted ferroptosis process which activates compensatory mitophagy survival pathways to impact epithelial cell phenotypes in asthma. This is interesting paper for the treatment strategy against severe asthma. The authors have previously shown that autophagy is an antagonist of ferroptosis under conditions of Type 2 inflammation/IL-13 high expression, through complex interactions between three proteins, 15LO1, PEBP1 and LC3, and one hydroperoxyl phospholipid, 15-HpETE-PE. This study is not novel in that it involves the same autophagy mechanism with differences between macro- and organelle-specific autophagy, and that damaged mitochondria are already known to be removed by mitophagy. However, I believe that the theme of this study, in which the proferroptotic mitochondrial peroxidation process is antagonized by mitophagy, could be an important complement to previous papers. The followings are several concerns and questions raised in relation to this study.

Major comments:

1. Under asthmatic conditions (Th2 high), it can be understood from this paper that the augmentation of mitophagy counteracts pro-ferroptotic alterations in mitochondria, preventing severe asthma through cellular death. Then, why do abnormal alterations in mitochondria and dysfunction in cellular cilia occur even when the antagonistic action is operative? Moreover, if the authors contend that the complement by mitophagy is not sufficient against the mitochondrial peroxidation process, then why is ferroptosis not induced?
2. The authors have demonstrated cilia damage in HAEC-derived ALI subjected to IL-13 stimulation and in airway epithelial cells from asthma patients, illustrating a correlation with the severity of asthma. However, it is questionable to what extent abnormalities in cilia derived from

mitochondrial anomalies contribute to the exacerbation of asthmatic conditions and the worsening of Type 2 inflammation.

3. The authors elucidate the correlation with clinical pathology by employing an ex-vivo model utilizing primary airway epithelial cells in an ALI model, as well as freshly brushed airway cells derived from patients with asthma. However, the absence of in vivo studies yields a sense of inadequacy to explain the close relation between research findings and clinicopathology.

4. The authors mimic asthmatic conditions by adding IL-13 to HAECs. Exposure of the airway epithelium to allergens such as House Dust Mite (HDM) serves as a trigger in the pathophysiology of asthma, and it has been reported that DNA damage in the airway epithelium can be induced by HDM. Is it anticipated that the results of this study would also occur with HDM stimulation to the epithelium?

Minor comments:

Fig. 2C. The location of the cilia is not clear. Please illustrate the position of the cilia.

Fig. 4E. pPARKIN and PARKIN lines are unclear and indistinguishable.

Fig. 4. The authors have conducted a blockade of the pro-ferroptotic process utilizing siALOX (Fig 4); however, to demonstrate the involvement of mitophagy, which is a novel element in this study, should the authors attempt the attenuation or overexpression of mitophagy (OPTN or PINK1-parkin)?

Fig. 4H and Fig. S4B. The authors state in the text, "positively correlated with 15LO1 ($Rho=0.47$, $p=0.04$) and LC3-II ($r=0.57$, $p=0.01$) (Fig. 4H and Fig. S4B)." Given that these results do not indicate a correlation with various indices of asthma, should they be termed as "correlation"? What about the correlation with respiratory function, FeNO, and eosinophil count? For instance, since FeNO is downstream of IL-13, it is anticipated to correlate with the relative values of factors related to the proferroptosis-mitophagy axis induced by IL-13.

Reviewer #4 (Remarks to the Author):

The study by Yamada et al. presents a novel compartmentalized ferroptosis pathway in mitochondria that is driven by lipoxygenase and is shown for primary cells as well as HAECs from study participants.

This present review is based on my expertise in lipidomics and thus does not critically evaluate the other parts. In my opinion, I am missing some smaller in-depth informations about which statistical test was applied for which dataset and how many participants/primary cell replicates were tested in each figure.

Please find below specific commentaries and critics:

Line 614 - the sentence is bit misleading. As I understood, you detected PLs (including CL) and the two eicosanoid-esterified PEs (15-HETE PE and 15-HpETE PE). From your sentence it is indicated that there were also other eicosanoid-esterified PLs. Did they show some interesting behavior? If not, rephrase the sentence.

Line 636 - Which deuterated phospholipids did you use as internal standards? Please indicate.

Figure 2 F - It would be helpful to indicate the used statistical test in the figures caption. Was IL-13 compared to CTL in this case? Please indicate. + it seems like formation of oxPEs in relation to cardiolipin content is not consistent in every participant sample used, is there an explanation?

Figure 2 F - Although self-explanatory, the abbreviations are not indicated somewhere except for HC. Please add. Why is so less participant data shown in this graph? As indicated from Table 1 and 2 there were at least 16 participants with MA and 15 with SA. For LC-MS analysis, were only 4 HCs, 3 MAs and 3 SAs tested? Please indicate.

Figure 2 G - Same here, it seems like that 3 low 15LO1 and 5 high 15LO1-protein content participants were measured. + from my understanding, this figure represents only the participants with a severe asthmatic condition, which is well described in the results, but not in the figure caption. To avoid lack of understanding, please add more information in the caption.

Minor comments:

- Additional information on MSMS fragmentation and RT of the two Oxylipins would be a nice addition to the supplementary part

REVIEWER COMMENTS

Reviewer #1 (Remarks to the Author):

The manuscript by Yamada et al., describes the induction of ferroptosis via IL-13-induced 15LO1 production that leads to increased mitophagy. The confocal microscopy images are interesting to provide visualization of the data. Providing some human data from asthmatics provides some translation of the in vitro work in HAECs.

Main Points:

1. Ferroptosis has been shown to have a role in severe asthma. Other than showing that IL-13 decreases GPX4, determination of ferroptosis was not measured - the availability of redox-active iron and the oxidation of PUFA-containing phospholipids. Is the degree of ferroptosis different in mild vs severe asthma? Does apoptosis actually occur?

Thank you for these comments. However, we believe it is important to note that IL-13 increases GXP4 (Figure 2A), suggesting an increase in its protective capabilities, but also supporting an increase in FPLs, which must be subsequently neutralized to prevent cell death. Consistent with this, in the original version of this manuscript, we reported increases in 15 HpETE-PE (the 15LO1 relevant PUFA-containing FPL) *ex vivo* in fresh asthmatic HAEC (Figures 2F-G), and consistent with our previous publication (*Wenzel et al Cell 2017*) where we also reported increases in 15 HpETE-PE in cultured HAECs in response to IL-13, and their reduction with 15LO1 KD. In the original version of the manuscript we added to this by showing increases in FPLs in *isolated mitochondrial fractions* following IL-13 stimulation and consistent with ferroptosis localized to mitochondrial compartments. Thus, in this current and in previous manuscripts. we have indeed shown specific increase in ferroptotic PUFA-containing ferroptotic phospholipids, both *in vitro* in whole cells and in mitochondrial fractions, as well as *ex vivo* in fresh epithelial cells. We have not addressed redox-active iron as that is beyond the current scope of the manuscript, although this should be pursued in the future. Finally, we have not observed initial evidence for increases in apoptosis under IL-13/T2-Hi conditions (caspase 3 cleavage) and believe that further investigation of apoptosis is beyond the scope of the current manuscript.

2. Mitophagy has been shown to have a role in severe asthma. An increase in PINK, Parkin, and LC3-II do not necessarily indicate an increase in mitophagy. Inhibition of mitophagy could show similar results because these markers will accumulate when they are not properly degraded. Need to measure autophagy activity (autophagy flux).

Thank you for these comments. We sincerely apologize that we were not aware of the studies of mitophagy in severe asthmatic fibroblasts (Ramakrishnan R PlosOne 2020). We have added a reference to this manuscript in the *Discussion (lines 481-482)*.

Although mitophagy has been previously reported in severe asthmatic fibroblasts (Ramakrishnan R PlosOne 2020), mitophagy in asthmatic epithelial cells has not been reported.

To address the 2nd part of this question, we point the reviewers to our previous report *in Zhao J et al PNAS 2020* in intact epithelial cells, which demonstrated an increase in LC3-II under HCQ conditions, which limits autophagic flux. In further specific response to this comment, we evaluated the expression of LC3-II under IL-13 alone and IL-13+HCQ conditions in mitochondrial fractions. As the reviewer can see in Figure 4D, HCQ treatment further increased LC3-II accumulation in mitochondria fractions. We added reference this in the results, lines 244-245 as follows:

HCQ pretreatment, which limits autophagic flux, further increased LC3-II accumulation in mitochondria fractions, supporting active autophagy (Figure 4D 2nd panel, and Supplementary Fig.4D).

3. Mitophagy is induced by mitochondrial dysfunction. The two major measures of dysfunction are increased mtROS and a reduction/loss in membrane potential. Does IL-13 or 15LO1 mediate these abnormalities?

We thank the reviewer for this excellent suggestion and agree that mitophagy is primarily driven by increases in mtROS and loss of membrane potential ($\Delta\Psi_m$). To determine whether mitochondria from primary HAECs stimulated with IL-13 or fresh epithelial cells from asthmatic patients exhibit signs of both increases in mtROS and loss of $\Delta\Psi_m$, two separate experiments were performed. First we evaluated $\Delta\Psi_m$ using flow cytometry and the TMRM dye which accumulates in a membrane potential-dependent way, under both control and IL-13 cultured conditions. As can be seen in NEW Figure 3A-B, IL-13 significantly lowered the Ψ_m as compared to control cells. Text was added as follows, Lines 198-204.

To determine whether IL-13 stimulated primary HAEC exhibit loss of $\Delta\Psi_m$, cells treated (or not) with IL-13 were evaluated by flow cytometry using the cell-permeant fluorescent dye, tetramethylrhodamine methyl ester (TMRM), that accumulates in mitochondria in a membrane potential-dependent way (Creed, 2019). Two cell subsets were identified: subset A having a higher level of TMRM fluorescence, and subset B characterized by lower TMRM fluorescence (Figure 3A). IL-13 stimulation decreased TMRM fluorescence compared to controls in both subsets, indicative of decreased $\Delta\Psi_m$. (Figure 3A-B).

We then addressed the presence of increased mtROS in fresh primary HAECs from asthmatic patients by addressing the percentage of cardiolipin (the phospholipid only found in mitochondrial membranes) present in the mono-lyso form. Mono-lyso CL represents a remodeled CL, remodeled because of ongoing and repeated oxidative insults (Horvath and Daum, 2013-see below for actual references). As can be seen in NEW Figure 3D fresh asthmatic HAECs demonstrated 2-4 fold higher levels of remodeled CL than control cells, consistent with repeated oxidative insults. Further, these remodeling changes correlated very strongly with lung function changes, as measured by FEV1% predicted ($r=0.90$, $p<0.0001$) Given the profound selectivity of CL for mitochondria (Claypool, 2009, Horvath and Daum 2013), we believe these findings in primary HAECs from asthmatic patients represent a new paradigm to address mitochondrial dysfunction in cells from human patients and therefore have added these findings and figures to the main Results, in a new section, Lines 204-218 and as follows (references follow)

Loss of $\Delta\Psi_m$ in vitro is supported ex vivo by measurements of CL, exclusively localized to the inner mitochondrial membrane (Claypool, 2009, Horvath and Daum 2013), and readily peroxidized by ROS generated in mitochondria (CLOx). Interestingly, the amounts of CLOx were very low in fresh primary HAECs ex vivo, without significant differences between healthy and asthmatic samples (Supplementary Fig. 3). Since CLOx is effectively hydrolyzed by group VIB Ca²⁺-independent phospholipase A2 γ (iPLA2 γ) yielding mono-lyso-CLs (mCLs) and oxygenated fatty acids as the major products (Liu et al., 2017, Tyurina et al., 2014), and iPLA2 γ is predominantly distributed to mitochondria (Hara et al., 2019), mCLs levels were measured in fresh primary HAECs as well. Figure 3C shows elevated levels of mCLs in Mild-Moderate (MOD) and severe asthma (SA) as compared to HC samples (Figure 3C), Importantly, levels of these mCLs correlated strongly with forced expiratory volume in 1 second (FEV1)% predicted ($r=0.9$,

p<0.0001) and with fraction exhaled nitric oxide (FeNO), a marker of Type-2 inflammation (Spearman's $Rho=0.7$, $p=0.047$). Thus, mCL likely represents a remodeling of mitochondrial phospholipids, which may contribute to and be a consequence to mitochondrial dysfunction (Chicco and Sparagna, 2007, Duncan, 2020)

Supporting References.

Claypool SM. Cardiolipin, a critical determinant of mitochondrial carrier protein assembly and function. Biochim Biophys Acta. 2009 Oct;1788(10):2059-68. doi: 10.1016/j.bbame.2009.04.020. Epub 2009 May 5. PMID: 19422785

Horvath SE, Daum G. Lipids of mitochondria. Prog Lipid Res. 2013 Oct;52(4):590-614. doi: 10.1016/j.plipres.2013.07.002. Epub 2013 Sep 2. PMID: 24007978

Duncan AL. Monolysocardiolipin (MLCL) interactions with mitochondrial membrane proteins. Biochem Soc Trans. 2020 Jun 30;48(3):993-1004. doi: 10.1042/BST20190932. PMID: 32453413

Tyurina YY, Poloyac SM, Tyurin VA, Kapralov AA, Jiang J, Anthony-muthu TS, Kapralova VI, Vikulina AS, Jung MY, Epperly MW, Mohammadyani D, Klein-Seetharaman J, Jackson TC, Kochanek PM, Pitt BR, Greenberger JS, Vladimirov YA, Bayir H, Kagan VE. A mitochondrial pathway for biosynthesis of lipid mediators. Nat Chem. 2014 Jun;6(6):542-52. doi: 10.1038/nchem.1924. Epub 2014 Apr 20. PMID: 24848241

Liu GY, Moon SH, Jenkins CM, Li M, Sims HF, Guan S, Gross RW. The phospholipase iPLA₂γ is a major mediator releasing oxidized aliphatic chains from cardiolipin, integrating mitochondrial bioenergetics and signaling. J Biol Chem. 2017 Jun 23;292(25):10672-10684. doi: 10.1074/jbc.M117.783068. Epub 2017 Apr 25. PMID: 28442572

Chicco AJ, Sparagna GC. Role of cardiolipin alterations in mitochondrial dysfunction and disease. Am J Physiol Cell Physiol 2007; 292: C33-C44 [PMID: 16899548 DOI: 1

Hara S, Yoda E, Sasaki Y, Nakatani Y, Kuwata H. Calcium-independent phospholipase A₂γ (iPLA₂γ) and its roles in cellular functions and diseases. Biochim Biophys Acta Mol Cell Biol Lipids. 2019 Jun;1864(6):861-868. doi: 10.1016/j.bbalip.2018.10.009. Epub 2018 Nov 1. PMID: 30391710 0.1152/ajpcell.00243.2006]

4. Most of the Western blots and confocal images are not quantified, so the reproducibility of the data is questionable.

Thank you. We have quantified all western blot data and confocal images, and included them in the supplemental figures. We would respectfully point out, however, that these densitometry data do not always show significance at $p<0.05$, despite near total consistency of direction across 3 distinct human sample replicates. As the reviewer is likely aware, WB is generally considered a poorly quantitative methodology, without standard curves. Further, our experiments also exclusively utilize heterogeneous primary human cultures, not cell lines. Thus, we are hopeful that reviewer will view the images of the WBs and agree with the consistency of the results.

5. Some controls are missing in several figures, such as Fig. 1G, Fig. 2C, Fig. 3C, 3F, & 3H, 4F, 4G, 4I, & 4J.

Thank you and we apologize for the confusion. The corresponding controls in each figure were actually present in the original manuscript. In Figure 1G, 3H, 4F and 4G, the comparisons are between ALOX15 KD (siALOX15 or BLX2477) to negative control condition (siNC or DMSO) all under IL-13 conditions. These controls are present in the figures. We did not perform ALOX15 KD or 15LO1 inhibition under non-IL-13 control conditions in most cases, as 15LO1 levels under control (non-IL-13) conditions are routinely very low or undetectable.

For the *ex vivo* staining in Figure 2C, 3C, 3F, 4I and 4J, we only showed representative data from asthmatic participants in our original submission. Upon the reviewer's request, we now include similar staining of healthy control cytopins, where we observe a trend towards more expression and co-localization in asthmatic samples. However, these IF images of dual proteins are not easily quantifiable and comparable from one batch to the next, and we have limited sample numbers available. We currently do not feel comfortable comparing these two groups in the small numbers available to us. Thus, given the limitations of the current data, we have elected to show data from representative participant-types, without quantitative comparison. We will need larger numbers of participants to do comprehensive quantitative analysis. As the reviewer can likely appreciate, the *ex vivo* portion of this study moves forward based on availability of human samples. These studies are clearly important and will be further addressed in the future.

6. How does 15LO1 get into mitochondria, and what mitochondrial compartment - OMM, IMS, IMM, or matrix - does it locate?

Thank you for this comment. We agree these are extremely interesting and important questions. However, these studies in primary human airway epithelial cells are challenging at best and currently beyond the scope of this manuscript, requiring extensive targeted investigations.

7. Because IL-13 appears to only effect a few cells in the monolayer - Fig. 2B, Fig. 3E & 3H - is the phenotype of the entire monolayer changed?

Despite the reviewer's comment, we are not convinced that IL-13 only affects a few cells in the multi-layered (not monolayer) ALI culture system, but, our study was clearly not designed to address this. While we agree that select cells demonstrate evidence for responses to IL-13 along our pathways of interest, IL-13 could be having additional effects on other cells. To the second point, there are abundant data which show that the overall cell phenotypes of the ALI cultures changes with IL-13, As has been reported numerous times, there is an increase in goblet cells and a loss of ciliated cells under IL-13 conditions. We have reported previously that inhibition or knockdown of 15 LO1 leads to a decrease in goblet cells, although the mechanisms for that loss were unclear (Zhao J Am J Resp Crit Care Med 2009, Am J Resp Cell Mol Biol 2017). We now believe (and report) that some of the effect on goblet cells may be indirect, through effects to decrease ciliated cell numbers. However, many additional studies would be needed to determine all the mechanisms (or cell types) which are affected by IL-13 and 15LO1.

8. How does ferroptosis, mitophagy, or 15LO1 change cilia number, length, or function?

Thank you for this important comment. We agree that this question is central to the relevance of our paper and have addressed this concern with additional experiments. The impact of ferroptosis on cilia cell numbers was examined by WB, IF/CF and SEM in our standard 14-day ALI culture model. We utilized both broad ferroptosis inhibition with FER-1 and specific 15LO1 inhibition with BLX2477. As the reviewer can see in NEW Figure 6C, inhibition of ferroptosis

(and its subsequent inhibition of mitophagy) leads to clear increases in ciliated cell numbers under IL-13 conditions, where normally there are very few ciliated cells. Additional text was added as follows, Lines 330-333:

Inhibition of ferroptosis with the 15LO1 inhibitor FER-1 or BLX2477 restored tubulin expression by IF/CF (Figure 6C/Supplementary Fig. 6B) and by WB (Figure 6D) and improved mature cilia development at 14 days in ALI culture (Figure 6C, SEM and IF/CF).

While determining the ultimate mechanism for this effect requires many additional studies, the data are quite clear that ferroptosis, loss of $\Delta\Psi_m$ and subsequent mitophagy inhibits the development of ciliated cells. Studies which address the mechanisms by which cilia length, numbers and function are affected are ongoing, and may involve worsening mitochondrial function and ATP generation. However, we believe these extensive additional metabolic studies are beyond the scope of the current paper.

9. Many of the conclusions in the figures are overstated or incorrect. For example, LC3-II is actually increased in Fig. 3G with siALOX, and COXII is present in the mitochondrial fraction with or without IL-13 in Fig. 2A.

We apologize for our presentation increased your confusion. Old Figure 3G (now Figure 4G) only included IL-13 conditions, comparing siNC (negative control) with siRNA KD of 15LO1 (siALOX15) in mitochondria fractions. We hope the reviewer agrees that siALOX15 decreases LC3-II as compared to siNC. We also hope the reviewer agrees that we should not compare the effects of siALOX15 between the cytosolic and mitochondria fractions as the protein concentration of the cytosolic fraction is much different from that of the mitochondria, especially after the extensive process of extraction and concentration.

For Figure 2A, we agree that COXII is represented in mitochondria fractions under both control and IL-13 conditions, as would be expected. However, there is a slight decrease in COXII when normalized to the loading control GAPDH (in the cytosolic fraction), consistent with other mitochondrial proteins such as those in the WB for Figure 4D.

Normalization for quantification by WB is often challenging, which is one of the reasons we attempt to present supporting data using several methodologies. However, we have added the following detail in the **Methods** section, Lines 697-704.

Quantification of proteins in cell compartments. *Normalization for quantification by WB is often challenging. As IL-13 stimulation decreases intracellular mitochondria numbers/volume, indexing to a mitochondrial marker, as a housekeeping gene, is not appropriate. Accordingly, to evaluate quantitative changes in mitochondrial proteins under IL-13 conditions, the entire mitochondria fraction obtained after separation was loaded onto the gels, each of which was derived from the original total cell protein (as defined by GAPDH levels). The proteins in the total mitochondrial fraction were then indexed to the starting total GAPDH in the cell (as measured in the cytosolic fraction), and as has been previously reported and performed.*⁸⁵⁻⁸⁹

10. Many of the Western blots are poor quality. There is a halo around the band.

We appreciate this comment and have done our best to improve the quality of the Western Blots. As the reviewer rightly points out, some of the bands had halos around them. We discovered this was because of magnification of lower resolution images. We apologize for this error. We now present high resolution images, which we believe considerably improve the overall

images. It is also important to note that for proper, scientifically rigorous, evaluation of mitophagy, it is critical to present single blots which have been probed multiple times for multiple targets (LC3, OPTN, PINK1, Parkin, pPARKIN, 15LO1, mitochondrial markers, etc) As we believe the reviewer likely understands, repeated probing and stripping leads to less distinct bands. All blots presented are from the same experiment, which we believe is the most rigorous presentation of the data.

Reviewer #2 (Remarks to the Author):

This study by Yamada et. al sought to define the mechanism by which 15 lipoxygenase-1 15LO1 (or ALOX15) which is highly expressed in the airway epithelium of individuals with asthma intersects with ferroptosis, focusing on mitochondria. The authors first show that IL-13 treatment of HAECs resulted in a reduction in the number and length of mitochondria. Upon an acute dose of the ferroptosis inducer RSL3, mitochondrial fragmentation was enhanced in IL-13-treated HAECs and prevented by dicer siRNA to ALOX15. Using mitochondrial enriched fractions, the authors then show that 15LO1 expression increases in both mitochondrial enriched and cytosolic fractions upon 1L-13 stimulation aligned with GPX4 expression. Immunofluorescence (IF) also showed some localization between 15LO1 and ATPsynthase. These IF findings were recapitulated in HAEC from individuals with asthma. Next HPLC/MS was used to measure oxidized lipids in mitochondrial enriched fractions upon IL-13 stimulation showing increased levels of 15OL1 oxidized PE species. Total oxidized PE's were also higher in whole cell lysates from HAECs from individuals with asthma. IL-13 stimulated autophagy, defined as increased LC3-II followed the induction of 15LO1 protein, and co-localization of these two proteins was observed by IF as well as in mitochondrial enriched fractions. Finally, the authors show that PINK1-Parkin as well as OPTN are activated in HAECs upon 1L-13 stimulation and that mitophagy membranes were observed in ciliated epithelial cells by TEM. Mitophagy markers were also higher in freshly brushed HAECs from asthmatics with enrichment in ciliated epithelial cells and this correlated with those cells that had shorter cilia- an effect that correlated with more severe asthmatic disease.

This work builds on prior published observations by this group that human airway epithelial cells (HAECs) release extracellular mtDNA release upon stimulation with inducers of ferroptosis in the presence of IL-13 and that the autophagy regulator LC3-II limits such an increase. This study is important in delineating the role of mitochondrial biology in airway epithelial cells in asthma pathogenesis. Strengths of this study include the use of primary human AECs grown at ALI, as well as fresh AECs from healthy controls and individuals with asthma with some clinical characterization. The use of subcellular HPLC/MS measurements in crude mitochondrial fractions is innovative. There is strong translational relevance, and the use of IL-13 ALI model appears robust. The link between cilia length/function and 15LO1 is novel and of interest.

Thank you for these positive comments.

Limitations to this study include the use of the method used to generate "mitochondrial" fractions, the lack of clarity surrounding the role of ferroptosis in IL-13-induced autophagy/mitophagy, some conceptual overlap with prior publications regarding the role of autophagy/mitophagy in controlling cilia length and the role of autophagy in asthma pathogenesis. There are three major proposed mechanisms (ferroptosis via 15LO1, autophagy/mitophagy, and cilia shortening/loss) with neither one explored robustly in detail.

Thank you for these comments. We hope our detailed responses to your concerns below more robustly detail the mechanisms governing these processes.

To expand on these concerns.

Major concerns.

1. The Abcam mitochondrial fractionation kit is a useful tool; however, the mitochondrial fraction produced may not be as pure as expected with peroxisome and lysosomes present (Curr Protoc Cell Biol 2001 May: Chapter 3:Unit 3.4) and is usually termed the “crude” mitochondrial fraction. Given the important role of lipids. (and possible ferroptosis) in both of these organelles, it is imperative that the authors prove there is no contamination in their fractions using organelle markers specific for peroxisomal proteins or lysosomal proteins or use an alternative method such as gradient density centrifugation or the generation of mitoplasts (if interested in the matrix). Given the reliance of the majority of findings on this method, it is hard to deduce how mitochondrial-specific these findings are.

Thank you for this excellent point. We too observed this impurity in our early experiments. After discussing with the manufacturer’s technical support, we modified the protocol by lowering the centrifugation speed to 6000g, but spinning for a longer time (15min). We then washed the pellet 2X with PBS. While this modification may decrease the total mitochondrial yield per isolation, it significantly improved mitochondrial purity. As can be seen in the WBs of cytosolic (GAPDH), mitochondria electron transport chain (ETC) (COXII/NDUFB8) and as suggested by the reviewer, mitochondrial membrane (TOM20) proteins (NEW Figure 2D), these markers are exclusively expressed in their corresponding fraction, supporting the purity of the mitochondria fraction. We added details of this modified protocol in the revised Methods section Lines 670-680:

Mitochondrial fractions were isolated using the Mitochondria Isolation Kit for Cultured Cells (ab110171, Abcam) with some modification of the manufacturer’s instructions. Briefly, HAECs were collected from transwell membranes with a cell lifter and pelleted by centrifugation at 700 g for 5minutes. The cells were frozen, thawed to weaken cell membranes, resuspended in Reagent A and then incubated on ice for 10 minutes. Cells were transferred into a pre-cooled Dounce Homogenizer for 3-5 strokes. Homogenates were centrifuged at 1,000 g for 10 minutes at 4°C and saved as supernatant #1. The pellet was resuspended in Reagent B and cell rupturing repeated.

However, given the phospholipid expertise of our laboratories, we went beyond simple protein validation, utilizing the diversity and specificity of phospholipids which are essential for the harmonized functioning of membranes in the cell and its organelles (Vance 2015). Indeed these phospholipids can also be used to differentiate cell compartments. For example, cardiolipins (CLs) are uniquely characteristic of mitochondria, particularly its inner membrane (Horvath and Daum, 2013). Vice versa, phosphatidylserines (PSs) are lacking from mitochondria (Leventis and Grinstein 2010). Based on these findings, the presence or absence of phospholipids in particular subcellular organelles has been effectively utilized for characterization of the purity of isolated subcellular fractions (Veyrat-Durebex et al., 2018; Brügger, 2014). With the advent of high-resolution soft-ionization mass-spectrometry, lipidomics features of cell organelles found applications as even more informative biomarkers (Hornemann, 2022; Bandu et al., 2018; Samhan-Arias et al., 2012). Predominance of individual molecular species of phospholipids was associated with membrane/organelle functions as exemplified by the role of bis(monoacylglycero)phosphatidic acid species (BMPs) in hydrolytic reactions in lysosomes (Schulze and Sandhoff, 2011). With this in mind, we utilized phospholipid-based evaluation of the purity of the two major fractions – mitochondria and cytosol - isolated from airway epithelial cells, focusing on four phospholipid classes as biomarkers of fractions’ purity. Because PS species are effectively decarboxylated to the respective phosphatidylethanolamines (PEs) in mitochondria, the abundance of PSs in mitochondria is very low (Vance 2015). Combined with the high abundance of CL in mitochondria, the CL/PS ratio represents a sensitive and specific marker

of mitochondrial purity. Indeed, in line with our protein measurements, our phospholipid assessments demonstrated a highly significant difference between the CL/PS ratios for mitochondrial and cytosolic fractions (NEW Supplemental Figure 7A). This is further supported by our estimated CL/BMP ratios as diagnostic of possible contamination of mitochondrial fraction by cytosolic lysosomes (NEW Supplemental Figure 7B). Our results further support the statistically significant predominance of CLs in mitochondria, along with the low amounts of BMPs (cytosolic, lysosomal) in these fractions.

To the reviewer's point regarding contamination with peroxisomes, biosynthesis of plasmalogens is associated with peroxisomes in the endoplasmic reticulum, ie in the cytosolic fractions of cells (Abe et al., 2020) Thus the cytosolic compartment should be enriched in plasmalogenic forms of PE whereas mitochondrial PE should be enriched with di-acylated-molecular species of PEs (Nagan and Zoeller, 2001). Based on this, we performed direct comparative LC-MS quantitative assessments of the major PE-plasmalogens (PEp(34:1); PEp(36:2); PEp(36:3); PEp(36:4); PEp(38:3); PEp(38:4); PEp(38:5) vs the levels of the corresponding diacylated PE-species (PEd) (PE(34:1); PE(36:2); PE(36:3); PE(36:4); PE(38:3); PE(38:4); PE(38:5). In this way, we employed specific phospholipid features to respond to the Reviewer's excellent recommendation to assess peroxisomal vs non-peroxisomal protein markers. We found that the cytosolic fraction was enriched in the PEp as evidenced by the highly significant differences in PEp/PEd ratios in the cytosol vs mitochondria, respectively. Thus, while we agree that the methodology used has the potential for impurities in the mitochondrial fraction, the fractionation appears to have been sufficiently pure to support the presence of FPLs in the selective mitochondrial fractions.

These new data are referred to in the results, presented in the **Methods** section and in NEW Supplementary Fig. 8 as follows:

Results (lines 153-57): *This relative purity was supported by measurement of compartment-specific lipids of interest, including cardiolipin (CL), phosphatidylserine (PS) (mitochondria vs cytosol specific) and bis(monoacylglycero)phosphatidic acid species (BMPs), as well as PE-plasmalogens (specific for lysosomes and peroxisomes) See Isolation of Mitochondria Fractions in Methods, and accompanying Supplementary Figures 8A-C for validation by specific lipids.*

Methods (Lines 682-695)

The mitochondrial and cytosolic fractions were evaluated for relative purity by measuring both compartmental lipids using LC-MS/MS and proteins (Western blot) of interest. Cardiolipins (CLs) are uniquely characteristic of mitochondria (Horvath and Daum, 2013), while phosphatidylserines (PSs) are absent (Leventis and Grinstein 2010). Additionally bis(monoacylglycero)phosphatidic acid species (BMPs) are specific to lysosomes (Schulze and Sandhoff, 2011). Thus, the CL/PS ratio is a sensitive and specific marker of mitochondrial purity. Supplementary Fig. 2A demonstrates highly significant differences between the CL/PS ratios in the mitochondrial and cytosolic fractions. The CL/BMP ratios (Supplementary Fig. 2B) further confirm limited contamination of mitochondrial fractions by cytosolic lysosomes (Supplementary Fig. 2B). To exclude potential contamination with peroxisomes in the mitochondrial fraction, we assessed the presence of plasmalogens (Abe et al., 2020), such that the cytosolic compartment would be enriched in plasmalogenic forms of PE whereas mitochondrial PE is enriched with di-acylated-molecular species of PEs (Nagan and Zoeller, 2001). Indeed, the cytosolic fraction was enriched in PEp as evidenced by the highly significant differences in PEp/PEd ratios in the cytosol vs mitochondria, respectively (Supplementary Fig. 2C). These data collectively support the relative purity of the mitochondria fractions.

Supplementary Figure 8A-C: Assessments of organelle-specific phospholipids to characterize the purity of mitochondrial and cytosolic fractions isolated from HAEC.

Ratio of CL/PS (left panel), CL/BMP (middle panel) and PEP/PEd (right panel) in the cytosolic (CT) and mitochondrial (MT) fractions. CL-cardiolipin, PS-phosphatidylserine, PEP-phosphatidylethanolamine plasmalogens, PEd-phosphatidylethanolamine diacyls, BMP-bis-(mono)glycerophosphate. Statistics by unpaired two-tail *t*-test.

Supporting References.

Abe Y, Honsho M, Kawaguchi R, Matsuzaki T, Ichiki Y, Fujitani M, Fujiwara K, Hirokane M, Oku M, Sakai Y, Yamashita T, Fujiki Y. A peroxisome deficiency-induced reductive cytosol state up-regulates the brain-derived neurotrophic factor pathway. J Biol Chem. 2020 Apr 17;295(16):5321-5334. doi: 10.1074/jbc.RA119.011989. Epub 2020 Mar 12. PMID: 32165495

Horvath SE, Daum G. Prog Lipid Res. Lipids of mitochondria. 2013 Oct;52(4):590-614. doi: 10.1016/j.plipres.2013.07.002. Epub 2013 Sep 2. PMID: 24007978

Nagan, N.; Zoeller, R.A. Plasmalogens: Biosynthesis and functions. Prog. Lipid Res. 2001, 40, 199–229.

Schulze, H.; Sandhoff, K. Lysosomal lipid storage diseases. Cold Spring Harb Perspect. Biol. 2011, 3.

Vance JE. Phospholipid synthesis and transport in mammalian cells. Traffic. 2015 Jan;16(1):1-18. doi: 10.1111/tra.12230. Epub 2014 Oct 15. PMID: 25243850

Leventis PA, Grinstein S. The distribution and function of phosphatidylserine in cellular membranes.

Annu Rev Biophys. 2010;39:407-27. doi: 10.1146/annurev.biophys.093008.131234. PMID: 20192774

Schulze, H.; Sandhoff, K. Lysosomal lipid storage diseases. Cold Spring Harb Perspect. Biol. 2011, 3.

Veyrat-Durebex C, Bocca C, Chupin S, Kouassi Nzougnet J, Simard G, Lenaers G, Reynier P, Blasco H. J Proteome Res. Metabolomics and Lipidomics Profiling of a Combined Mitochondrial Plus Endoplasmic Reticulum Fraction of Human Fibroblasts: A Robust Tool for Clinical Studies. 2018 Jan 5;17(1):745-750. doi: 10.1021/acs.jproteome.7b00637. Epub 2017 Nov 14. PMID: 29111762

Brügger B. Lipidomics: analysis of the lipid composition of cells and subcellular organelles by electrospray ionization mass spectrometry. Annu Rev Biochem. 2014;83:79-98. doi: 10.1146/annurev-biochem-060713-035324. Epub 2014 Mar 3. PMID: 24606142

Hornemann T. Lipidomics in Biomarker Research. Handb Exp Pharmacol. 2022;270:493-510. doi: 10.1007/164_2021_517. PMID: 34409495

Bandu R, Mok HJ, Kim KP. Phospholipids as cancer biomarkers: Mass spectrometry-based analysis. Mass Spectrom Rev. 2018 Mar;37(2):107-138. doi: 10.1002/mas.21510. Epub 2016 Jun 8. PMID: 27276657

Samhan-Arias AK, Ji J, Demidova OM, Sparvero LJ, Feng W, Tyurin V, Tyurina YY, Epperly MW, Shvedova AA, Greenberger JS, Bayır H, Kagan VE, Amoscato AA. Oxidized phospholipids as biomarkers of tissue and cell damage with a focus on cardiolipin. Biochim Biophys Acta. 2012

2. The use of the low dose acute treatment of RSL3 in Figure 1 is used to claim that compartmentalized ferroptosis may occur with loss of ALOX15 in HAECs reducing the effect of IL-13/RSL3 on mitochondrial numbers structure and autophagy. Yet for the rest of the manuscript, RSL3 is not used and other classic inducers of ferroptosis or inhibiting ferroptosis by classical means are not linked directly to il-13-induced mitophagy/autophagy. Can ferroptosis inhibitors (or lowering iron) protect HAECs from il-13-induced mitophagy, cilia shortening/loss etc.

Thank you for your comment. We use RSL3 in Figure 1 primarily as an “acute/short term” stimulus which accentuates the effects of ferroptosis on mitochondria fragmentation/numbers by increasing levels of 15 HpETE-PE. We also previously showed (Zhao J et al PNAS 2020) that RSL3 treatment of HAECs increased activation of LC3 (LC3-II) supporting a link between ferroptosis *activators* and autophagy/mitophagy. We performed additional longer term culture experiments with the addition of ferrostatin-1 (FER-1, a ferroptosis inhibitor). Similar to 15LO1 inhibition or knockdown, the studies demonstrate a reduction in autophagy/mitophagic markers (OPTN, LC3-II), in NEW Figure 5D following FER-1 treatment. We added the following text, Lines 288-293:

To confirm that 15LO1 mediated ferroptotic processes drove PINK1-Parkin-OPTN executed mitophagy, the ferroptosis inhibitor FER-1 and the 15LO1 inhibitor BLX2477 were added simultaneous with IL-13. As shown in Figure 5D and Supplementary Fig. 5H-I, both FER-1 and BLX2477 suppressed IL-13-induced OPTN and LC3-II expression. Additionally, BLX2477 decreased OPTN-LC3 colocalization (Figure 5E and Supplementary Fig. 5J-K).

Importantly, as we previously responded to Reviewer 1, we now also show that both 15LO1 inhibition and Fer-1 treatment increase the numbers of ciliated cells by Western Blot (acetylated α -tubulin), IF/CF and by scanning EM under IL-13 conditions. We added text in the Results, Lines 330-333 as follows:

Inhibition of ferroptosis with the 15LO1 inhibitor FER-1 or BLX2477 restored tubulin expression by IF/CF (Figure 6C/Supplementary Fig. 6B) and by WB (Figure 6D) and improved mature cilia development at 14 day ALI cultures (Figure 6C, SEM and IF/CF).

3. Lack of readouts of mitochondrial function such as membrane potential, respiration, movement etc.

Thank you for this suggestion. Please see our response to Reviewer #1, Comment #3. These new data confirm loss of Ψ_m in the presence of IL-13 as compared to control conditions. These findings were mirrored in fresh epithelial cells by showing the several fold change in mono-lyso CL in asthmatic samples, consistent with ongoing mitochondrial oxidative stress and mitochondrial phospholipid remodeling. We are quite encouraged by the very strong relationship of these ex vivo findings to lung function (FEV1% predicted), with an r-value of 0.9, and a p-value of <0.0001.

4. Which cell type is important? It is clear that the ciliated epithelial cell is the focus of this study, but what proportion of HAEC cultures at ALI are ciliated, goblet and basal remains elusive- i.e., the key experiments that demonstrate co-localisation of 15LO1 in mitochondrial fractions – is this in ciliated cells only? Or does this occur in goblet and basal cells too?

We believe it is likely that 15LO1 is present and active in nearly all T2 stimulated HAECs. While our evolving scRNAseq (unpublished because of small numbers) data support transcripts for 15LO1 in all these cells, considerable additional study is needed to determine whether there is ferroptotic phospholipid generating activity in each cell type. We previously published and have preliminary data that 15LO1-PEBP1 and ferroptosis are active in basal cells (Li Z, J Allergy Clin Immunol 2020), where the activity appears to lead to increases in CCL26/Eotaxin-3 expression. Additional submitted/unpublished data from Dr. Kagan's laboratory demonstrate 15LO1 in basal cells of psoriatic keratinocytes, where the activity also appears to contribute to inflammatory responses. Understanding the presence and activity of 15LO1 in each of the epithelial cell subtypes is of critical importance but beyond the scope of this study .

Further, our time course studies demonstrate that 15LO1-PEBP1 increases very early (Days 2-3) in ALI cell culture systems, before the development of a large population of ciliated cells (old Figure 4A). In fact, we would propose that this early expression/activity (as seen in our 7 day ALI culture system) is primarily occurring in basal cells, where the ferroptosis induced loss of mitochondria prevents cilia development and function. We now present additional early time course data for the development of ciliated cells, both with and without 15LO1 inhibition and with FER-1, which support an effect of ferroptosis on cell trajectories at a very early stage, ie, likely at the level of the basal cell. We have added these findings into the Results, and in NEW Supplemental Figures 6C-D and expanded on the comment in the Discussion to include likely activity in basal cells.

Results (lines 334-340): To determine whether some of these effects could be initiated at a very early stage, through effects of ferroptosis on basal cells, the progenitor cells required for all differentiated HAECs, an early ALI culture model, starting at Day 0 of ALI, was utilized where basal cells predominate. IL-13 treatment starting at ALI day 0 almost completely inhibited cilia development, which was increased by BLX277 and FER-1 determined by IF/CF and SEM (Figures S6C-E). These results suggest that ferroptosis could initiate its effects at an early stage of cell differentiation..

Discussion (lines 451-454): It is conceivable that expression/activity in basal cells induces mitochondrial loss limiting cilia development and function while promoting goblet cell differentiation. This concept is supported by the impact of ferroptosis on early cell differentiation where basal cells predominate.

We know very little about 15LO1/ferroptosis in goblet cells but strongly agree that would be an excellent area to explore in the future.

The role for mitochondria in each of these diverse cell types is very different; so, is this proposed pathway also specific to one cell type or all?

Please see the discussion above. Understanding the relationship of 15LO1 to mitochondria in each of these cell types is clearly important and worthy of further study, but we believe this is beyond the scope of the current manuscript.

5. Conceptually the link between IL-13 and autophagy has been described (Thorax. 2020 Sep;75(9):717-724) with autophagy essential for airway secretion of goblet cells in response to il-13 (Autophagy. 2016 Feb; 12(2): 397-409.). Cilia length and function in the airway epithelium has also been linked to autophagy (J. Clin. Invest. 123, 5212-5230 10.1172/JCI69636), and increased autophagy-related 5 gene expression is associated with collagen expression in the airways of refractory asthmatics (Front. Immunol. 8, 355 10.3389/fimmu.2017.00355).

Thank you for pointing these out. We agree that previous studies have linked autophagy to Type-2 inflammation and to mucus (including our own paper, *Zhao J et al PNAS 2020*). We are unaware of any previous link of T2/IL13 associated autophagy with the ciliated cell phenotype. We have added the reference for these papers, including highlighting the differences between these papers and our own in the Discussion. Specifically, results from the first paper (Thorax 2020) differ from the ones reported here, suggesting that IL-13 induced autophagy *increases* release of mitochondrial DNA (at extremely low 2-6 pg/ml levels), enhancing inflammation. Our result suggest the opposite, that autophagy/mitophagy is cell protective and limits mtDNA release, as we published previously (*Zhao J et al PNAS 2020*, Figure 5E). The reasons for these differences are unclear but may include methodologic differences, including our activation of ferroptosis which led to 3-log increases in released mtDNA, which was further increased by LC3 KD. The *Autophagy 2016* paper was primarily observational without mechanistic insights, but clearly relevant to the current manuscript. The relation of autophagy (not mitophagy) to cilia function in the JCI paper in COPD reports on a different disease, primarily in a mouse cigarette smoke model, with a profoundly different mechanism and did not link the responses to mitochondria or mitophagy in any way. Thus, although clearly important to cite, we believe that our current paper adds substantially to what is known. We do not feel that the Front Immunol paper, which analyzed biopsies/submucosal tissue, and without clear relation to severity of disease is less relevant to our manuscript and we would prefer not to add that reference.

We added the following text to the discussion (including the references).

Thorax 2020 (lines 404-407): To that end, this localized 15LO1 dependent ferroptotic mitochondrial disruption elegantly coordinates with lipidation of the quintessential autophagic membrane protein LC3 through shared PEBP1 activation pathways¹¹ to drive formation of mitophagic membranes and, in contrast to one previous study, limit cell death.

Autophagy 2016 (lines 459-461): Interestingly, IL-13 has also been reported to enhance mucus secretion from goblet cells in an autophagic dependent process,, which could further increase ciliated cell stress.

JCI 2013 (lines 482-484): Autophagy has also been previously linked to cilia dysfunction in chronic obstructive pulmonary disease, (COPD), particularly in relation to smoke induced oxidative stress, but the relation to human disease is less clear⁷³.

Minor concerns.

1. *The immunoblots presented using these mitochondrial enriched fractions should also include a mitochondrial housekeeping protein in addition to proteins like ETC proteins. i.e. Figure 2A- need tim23 or tom 20 loading control*

Per this Reviewer's request, we added TOM20 as the mitochondrial housekeeping protein in Figure 3D to further indicate the loading and purity of the mitochondria fraction. In fact, TOM20 appears very similar to the ETC protein markers.

2. *The authors mention in Figure 1A and Supplementary Fig.1A and B that mitochondria volume and size under both short (identified by TOM20) and longer (identified by ATPsynthase) term cultures are used- it is unclear why use two separate markers – clarification would be helpful here.*

This was purely related to antibody availability. Our TOM20 antibody Abcam (Cambridge, MA) was not available during the times we were performing these studies, at least partially due to COVID19 supply chain issues. However, we are hopeful the reviewer will appreciate that we were able to show very similar results at both a different time point and using a different mitochondrial marker.

3. Figure 1E and F- no RSL3 only controls.

Thank you for this suggestion. We agree and added supplemental data (n=3) and a figure showing the lack of an effect of RSL3 on HAECs under control conditions. We refer to these new data on Lines 124-126 and in NEW Supplemental Figure 1SD-F .

RSL3 treatment of control cells (with low 15LO1 expression) had little effect on mitochondria number, size and fragmentation/sphericity (Supplementary Fig. 1D-F), supporting the selectivity of these effects to conditions in which 15 LO1 is elevated.

4. mtDNA measurements not normalized to total nuclear DNA – so unclear of loss of mtDNA due to loss in cells

We normalized the mtDNA to total protein, similar to the use of GAPDH as loading control in WB. Support for an overall decrease in mtDNA also comes from IF staining (ATP synthase and TOM20) which was indexed to cell number for quantification. There are no remaining samples from these particular experiment to measure nuclear DNA assay. However, to address the Reviewer's concern about cell loss, we added LDH release data and WB of Histidine from new experiments (NEW Supplemental Figure 1C). Both LDH and Histidine WB data show that IL-13 did not induce cell death or loss of nuclear DNA, additionally supporting the conclusion that mtDNA loss is not due to overall cell death and loss. We add reference to these data in the Results, Lines 110-112 as below.

This loss of mitochondria occurred in the absence of greater cell death under IL-13 conditions (measured by LDH release) and without concurrent loss of nuclear DNA-associated material (histidine levels by western blot (WB))(Supplementary Fig. 1C).

5. Lack of quantification of TEM images. Thank you for the comments. After thoroughly considering this reviewer's comments and the set of data we are presenting, we believe it is unnecessary and even unhelpful and misleading to quantify the TEM images for the following reasons. First, TEM quantitation suffers from major under-sampling issues (70 nm section, few cells in the sections). There will always be issues with the numbers of cells that are available for imaging. In order to do TEM quantitation, a large number of cells, sections and samples are needed with intensive labor involved, which is usually not feasible or even reasonable for human samples. Second, we have done quantitation analyses on the cells using IF, and sometimes by WB, which are much more conclusive in that more cells and signals can be extracted and quantified. We would like to emphasize that the TEMs were done to *support* the IF (and WB) results in a more detailed/higher power view. We hope the reviewer will appreciate our efforts in this regard.

6. Scale bars lacking on all images. Added, thank you.

7. It is unclear how many biological and/or technical replicates are used in all panels of the figures. Thank you, we have now added these data to the figure legends or figures.

Reviewer #3 (Remarks to the Author):

Yamada et al investigated A 15-Lipoxygenase-1 dependent pathway drives a mitochondrially-

targeted ferroptosis process which activates compensatory mitophagy survival pathways to impact epithelial cell phenotypes in asthma. This is interesting paper for the treatment strategy against severe asthma. The authors have previously shown that autophagy is an antagonist of ferroptosis under conditions of Type 2 inflammation/IL-13 high expression, through complex interactions between three proteins, 15LO1, PEBP1 and LC3, and one hydroperoxyl phospholipid, 15-HpETE-PE. This study is not novel in that it involves the same autophagy mechanism with differences between macro- and organelle-specific autophagy, and that damaged mitochondria are already known to be removed by mitophagy. However, I believe that the theme of this study, in which the proferroptotic mitochondrial peroxidation process is antagonized by mitophagy, could be an important complement to previous papers.

Thank you for those overall positive comments.

The followings are several concerns and questions raised in relation to this study.

Major comments:

1. Under asthmatic conditions (Th2 high), it can be understood from this paper that the augmentation of mitophagy counteracts pro-ferroptotic alterations in mitochondria, preventing severe asthma through cellular death. Then, why do abnormal alterations in mitochondria and dysfunction in cellular cilia occur even when the antagonistic action is operative?

Thank you for that very important comment. We agree that the relationships are complex, but your 1st sentence summary of the findings is clearly in-line with our understanding. We certainly do not know all the details of these processes, particularly all the dynamics of ferroptosis initiation and mitophagic response, whether these processes are continuously ongoing or whether certain stimuli serve to initiate the ferroptosis or even the efficacy/efficiency of mitophagy to preserve mitochondrial components. In fact, autophagy/mitophagy almost always occur under conditions of cell stress, and as a response to that stress. Yet, that response may or may not be adequate to “save” the cell or certainly reverse the ongoing stress. Indeed, in our system, the mitophagy does not appear to reverse the ongoing T2/15LO1 activity and accompanying mitochondrial damage. Indeed, in our primary freshly brushed HAECs from asthmatic patients there is evidence that mitophagy does not protect the mitochondria from oxidative damage given the large percentage of remodeled CL relative to total CL, in asthmatic compared to healthy cells. While your point is indeed excellent, at this stage of our understanding we would posit that this compensatory mitophagy is only partially effective in preserving mitochondria, and preserving cilia numbers as well as their function in the face of ongoing ferroptosis. We look forward to further exploring the consequences of these damaged/remodeled mitochondria further but believe that will require extensive study outside of the scope of the manuscript.

Moreover, if the authors contend that the complement by mitophagy is not sufficient against the mitochondrial peroxidation process, then why is ferroptosis not induced?

Again, a very important point. However, we believe our data in Figures 1 and 2 *all support* the induction of ferroptosis, and ferroptotic lipid generation. Ferroptosis is defined by the generation of hydroperoxy-phospholipids which contribute to cell death. Increases in ferroptosis are observed following genetic or pharmacologic inhibition of glutathione peroxidase-4 (GPX4) while tailored anti-oxidants like ferristatin-1 (Fer-1) decrease ferroptosis. No proteins or cell morphologic changes specifically define ferroptosis. To this end, in the original manuscript we confirmed the generation of the ferroptotic phospholipid, 15 hydroperoxyeicosatetraenoic acid-phosphatidylethanolamine (15 HpETE-PE) by 15 lipoxygenase-1 (15LO1) in fresh human airway epithelial cells (HAECs) and their higher levels in asthmatic patients (Figure 2F). This

is consistent with and expands on the results presented previously in smaller numbers in our *Cell* 2017 paper. In this current manuscript, we expand on these findings by demonstrating the presence of these ferroptotic phospholipids (FPLs) in isolated mitochondrial fractions and show that these FPLs increase under T2/IL-13 conditions.

Specifically, there is mitochondrial generation of ferroptotic phospholipids following IL-13 stimulation, with a likely compensatory induction of GPX4 in response to the increase in these oxidized phospholipids (Figure 2A). The effects on mitochondria numbers/fragments are further enhanced by *an inducer* of additional ferroptosis (RSL3 which inhibits GPX4) (Figures 1B and D), supporting its induction, an induction not seen in control cells with low 15LO1 levels. Finally, inhibition of ferroptosis with FER-1 or KD/inhibition of 15 LO1 all reverse the mitochondrial effects. Overall, we believe this process represents a form of ferroptosis which involves mitochondrial membranes, and does not lead to cell death until either overwhelming loss of mitochondria occurs [particularly in the absence of autophagy/mitophagy (Zhao J et al PNAS 2020)] or when/if ferroptosis becomes more generalized.

2. The authors have demonstrated cilia damage in HAEC-derived ALI subjected to IL-13 stimulation and in airway epithelial cells from asthma patients, illustrating a correlation with the severity of asthma. However, it is questionable to what extent abnormalities in cilia derived from mitochondrial anomalies contribute to the exacerbation of asthmatic conditions and the worsening of Type 2 inflammation.

Thank you for this comment. It has long been known that ciliary dysfunction drives airway diseases, with the most clear example being that of primary ciliary dyskinesia (PCD), where the absence or reduction of ciliary beating because of genetic abnormalities in the dynein motors drives cilia dysfunction. PCD has been linked to development of bronchiectasis at an early age. We believe our data support that the abnormalities in cilia number/density under T2/IL-13 conditions that we report are similar (albeit to a lesser degree), than what is observed with PCD (Original Figure 5).

We are not, however, proposing that these *cilia abnormalities* contribute to worsened T2 inflammation. Indeed, we have reported several times that 15LO1-PEBP1 ferroptosis pathways contribute to enhanced T2-protein responses to IL-13 (Zhao J *Am J Resp Crit Care Med* 2009, Zhao J et al *PNAS* 2011, Zhao J *Am J Resp Cell Mol Biol* 2016, Li Z J *Allergy Clin Invest* 2020, Nagasaki T J *Clin Invest* 2022). However, we are not proposing (at least at this point) that these T2 inflammation-enhancing effects are through the impact on ciliated cells, but rather on basal or secretory cells. Clearly, understanding the cell sources of epithelial T2-signature proteins is relevant and worthy of further study, but it is not the focus of this manuscript.

3. The authors elucidate the correlation with clinical pathology by employing an ex-vivo model utilizing primary airway epithelial cells in an ALI model, as well as freshly brushed airway cells derived from patients with asthma. However, the absence of in vivo studies yields a sense of inadequacy to explain the close relation between research findings and clinicopathology.

We certainly agree that *in vivo* studies could enhance the cause-effect relationships! However, current mouse models for evaluation of epithelial mechanisms of relevance to humans are generally appreciated to be of modest help. The mouse airway epithelium is structurally dissimilar to human epithelia, with a cuboidal (as opposed to stratified columnar) epithelium. Gene expression profiles are different and the 12/15 lipoxygenase in mouse epithelial cells is quite dissimilar to the human 15LO1 enzyme. Mouse epithelial cells express extremely low levels of inducible nitric oxide synthase, even under T2- conditions, which is vastly different from

the profound upregulation of this pathway by IL-4/13 in humans. Thus, results from those experiments would likely be difficult to interpret. Indeed, we are hopeful that the reviewer will appreciate that the parallel findings in fresh primary HAECs from asthmatic patients as compared to cultured cells under IL-13 conditions (across all figures) give reassurance that our mechanistic studies reasonably model elements of human disease.

4. The authors mimic asthmatic conditions by adding IL-13 to HAECs. Exposure of the airway epithelium to allergens such as House Dust Mite (HDM) serves as a trigger in the pathophysiology of asthma, and it has been reported that DNA damage in the airway epithelium can be induced by HDM. Is it anticipated that the results of this study would also occur with HDM stimulation to the epithelium?

Thank you for this comment. Although we are uncertain whether HDM would induce the same changes, we feel overall it is unlikely. HDM stimulation does not mimic IL-13 stimulation of HAECs, or induce 15LO1 upregulation that we are aware of. Without 15LO1, targeted mitochondrial ferroptosis cannot occur (Please see Supplementary Fig. 1D-F and response to Reviewer 2). While non-enzymatic ferroptosis is certainly feasible, we hope the reviewer will agree that studying those processes *in vitro* are beyond the scope of this manuscript.

Minor comments:

Fig. 2C. The location of the cilia is not clear. Please illustrate the position of the cilia. We apologize for the confusion. White-arrows have been added to indicate the cilia.

Fig. 4E. pPARKIN and PARKIN lines are unclear and indistinguishable. Thank you. We have improved all the western blot images to high resolution. We apologize for not recognizing this deficiency with the initial submission and hope that the higher resolution images are more acceptable. However, as we pointed out to Reviewer #1, for proper, scientifically rigorous, evaluation of mitophagy we believe it is critical to present single blots which have been probed multiple times for multiple targets of relevance (LC3, OPTN, PINK1, Parkin, pPARKIN, 15LO1, mitochondrial markers, etc) As we believe the reviewer likely understands, repeated probing and stripping leads to less distinct bands. All blots presented are from the same experiment, which we believe is the most rigorous presentation of the data.

Fig. 4. The authors have conducted a blockade of the pro-ferroptotic process utilizing siALOX (Fig 4); however, to demonstrate the involvement of mitophagy, which is a novel element in this study, should the authors attempt the attenuation or overexpression of mitophagy (OPTN or PINK1-parkin)?

We evaluated the effect of siRNA knockdown of OPTN on RSL3-induced ferroptosis. The new data show that OPTN partially KD increases RSL3-induced cell death (NEW Supplemental Figure E). These data are similar to our previous results with LC3-II knockdown which led to an increase in RSL3-induced cell death (Zhao J et al PNAS 2020). We have added reference to these new experiments in the Results, Lines 297-299 as follows:

Similar to our previous findings with LC3 KD (Zhao et al PNAS 2020), siOPTN KD enhanced RSL3-induced LDH release (Supplementary Fig. 5H) confirming its functional mitophagic protective effect ferroptotic cell.

Fig. 4H and Fig. S4B. The authors state in the text, "positively correlated with 15LO1 (Rho=0.47, p=0.04) and LC3-II (r=0.57, p=0.01) (Fig. 4H and Fig. S4B)." Given that these

results do not indicate a correlation with various indices of asthma, should they be termed as "correlation"?

We are sorry for the confusion. These “correlative” studies are specifically meant to relate the expression of the optineurin (OPTN) mitophagy protein with 15LO1 and LC3 in fresh HAECs where mechanistic studies are currently impossible. Thus, we believe that term correlation is appropriate.

What about the correlation with respiratory function, FeNO, and eosinophil count? For instance, since FeNO is downstream of IL-13, it is anticipated to correlate with the relative values of factors related to the proferroptosis-mitophagy axis induced by IL-13.

Thank you for this suggestion. We added the correlation analysis of FeNO with LC3-II and OPTN to the text. Although FeNO positively correlates with 15LO1 ($r=0.59$, $p=0.0004$), this strong correlation has been reported previously. We added the marginal correlations of FeNO with OPTN ($\rho=0.37$, $p=0.066$) and LC3-II ($\rho=0.47$, $p=0.007$). The correlation of 15LO1 with FEV1 (as well as asthma control) was reported in the original submission. We modified the FEV1 results slightly as convention has now changed (per ATS/ERS guidelines) to report FEV1 as non-race corrected % predicted values. There were no substantive changes to the results, although the mediation of no/short ciliated cells between 15LO1 and FEV1% predicted became slightly stronger.

Reviewer #4 (Remarks to the Author):

The study by Yamada et al. presents a novel compartmentalized ferroptosis pathway in mitochondria that is driven by lipoxygenase and is shown for primary cells as well as HAECs from study participants.

This present review is based on my expertise in lipidomics and thus does not critically evaluate the other parts.

In my opinion, I am missing some smaller in-depth informations about which statistical test was applied for which dataset and how many participants/primary cell replicates were tested in each figure.

Thank you, we have added this information to all the figure legends and/or text.

Please find below specific commentaries and critics:

Line 614 - the sentence is bit misleading. As I understood, you detected PLs (including CL) and the two eicosanoid-esterified PEs (15-HETE PE and 15-HpETE PE). From your sentence it is indicated that there were also other eicosanoid-esterified PLs. Did they show some interesting behavior? If not, rephrase the sentence.

Phospholipids and their hydroxy and hydroperoxy derivatives, including 15-HETE-PE and 15-HpETE-PE, were analyzed by LC/MS using a Dionex Ultimate 3000 HPLC system coupled on-line to an Orbitrap Fusion Lumos mass spectrometer (Thermo Fisher Scientific) using a normal phase column (Luna 3 μm silica C18(2) 100 Å, 150 \times 2.0 mm, Phenomenex).

In this study, we focused on asthma associated changes in two major classes of phospholipids - cardiolipin (CL) and phosphatidylethanolamine (PE) – specifically related to mitochondrial dysfunction and ferroptosis, respectively. Thus, LC/MS analysis of CL and PE as well as their

oxygenated metabolites (hydroxy- and hydroperoxy-derivatives) has been performed. The sentence in the Methods section has been rephrased as follows, lines 633-638:

As CL and PE are specifically related to mitochondrial dysfunction and ferroptosis, respectively, their polyunsaturated species as well as oxygenated metabolites (hydroxy- and hydroperoxy-derivatives) were analyzed by LC/MS. A Dionex Ultimate 3000 HPLC system (normal phase column, Luna 3 μm silica C18(2) 100 Å, 150 × 2.0 mm, Phenomenex) coupled on-line to an Orbitrap Fusion Lumos (ThermoFisher Scientific) mass spectrometer was employed.

Line 636 - Which deuterated phospholipids did you use as internal standards? Please indicate.

As requested by the Reviewer we have indicated the deuterated phospholipids used as internal standards as follows, lines 658-665:

A mixture of deuterated phospholipids consisting of 1-hexadecanoyl(d31)-2-(9Z-octadecenoyl)-sn-glycero-3-phospho-ethanolamine (PE(16:0D31/18:1)), 1-hexadecanoyl(d31)-2-(9Z-octadecenoyl)-sn-glycero-3-phosphocholine (PC(16:0D31/18:1)), 1-hexadecanoyl(d31)-2-(9Z-octadecenoyl)-sn-glycero-3-phosphoserine (PS(16:0D31/18:1)), 1-hexadecanoyl(d31)-2-(9Z-octadecenoyl)-sn-glycero-3-phosphate (PA(16:0D31/18:1)), 1-hexadecanoyl(d31)-2-(9Z-octadecenoyl)-sn-glycero-3-phosphoglycerol (PG(16:0D31/18:1)), 1-hexadecanoyl(d31)-2-(9Z-octadecenoyl)-sn-glycero-3-phospho-(1'-myo-inositol) (PI(16:0D31/18:1)) (Avanti Polar Lipids) was used as internal standards. 1,1',2,2'-tetramyristoyl-cardiolipin (sodium salt) (Avanti Polar Lipids) was used as cardiolipin internal standard.

Figure 2 F - It would be helpful to indicate the used statistical test in the figures caption. Was IL-13 compared to CTL in this case? Please indicate. + it seems like formation of oxPEs in relation to cardiolipin content is not consistent in every participant sample used, is there an explanation?

Thank you. Figure 2F is ferroptotic phospholipid data from freshly brushing asthmatic and healthy control airway epithelial cells. Thus, there is no comparison by treatment, but only by participant groups (one-way ANOVA). We have added information on these statistical tests to the figure captions.

And secondly, yes, indeed there are variations within participant samples as the reviewer pointed out. This is to be expected with research which involves highly heterogeneous human samples. However, the overall trends remain significant despite this heterogeneity, robustly supporting the findings.

Figure 2 F - Although self-explanatory, the abbreviations are not indicated somewhere except for HC. Please add. Thank you. We have gone through the manuscript and made certain that all the abbreviations are appropriately defined in the text.

Why is so less participant data shown in this graph? As indicated from Table 1 and 2 there were at least 16 participants with MA and 15 with SA. For LC-MS analysis, were only 4 HCs, 3 MAs and 3 SAs tested? Please indicate.

We apologize for the confusion. Old Tables 1 (new Table 2) lists all the participants for the Western Blot studies and additional *in vitro* experiments. Table 3 specifically lists all the participants who contributed samples for the studies in Figure 6 which compare ciliated cell phenotype with 15LO1 and lung function. Given the large numbers of additional experiments, we have revised Table 2 to reflect the new participants. We have also gone through the

manuscript text and figure legends and added the specific “n” for each experiment. With this revision, and in response to other comments, we added “n” for several experiments, including the LC/MS/MS experiments and have updated Tables 1 and 2 appropriately.

Figure 2 G - Same here, it seems like that 3 low 15LO1 and 5 high 15LO1-protein content participants were measured. + from my understanding, this figure represents only the participants with a severe asthmatic condition, which is well described in the results, but not in the figure caption. To avoid lack of understanding, please add more information in the caption.

Thank you, this figure actually includes mild to severe asthma. We have added the following to the figure legend:

...and higher in those participants (mild to severe asthma) with high 15LO1 protein (by WB) using paired t-testing.

Minor comments:

- Additional information on MS/MS fragmentation and RT of the two Oxylipins would be a nice addition to the supplementary part

Thank you for this suggestion. In response to the Reviewer’s comment, in the revised manuscript we have presented MS/MS fragmentation analysis of two oxygenated metabolites of stearoyl-arachidonoyl-PE – 15-HETE-PE and 15-HpETE-PE. We have included the results in the revised manuscript as a NEW Supplementary Fig. 7A-B, which is referred to in the Methods, Lines 658-666

Supplementary Figure 7: MS/MS identification of 15-HETE-PE and 15-HpETE-PE species in mitochondrial fraction isolated from cells exposed to IL13.

A. MS/MS spectrum of PE molecular ion with m/z 782.538 at RT = 80.67 min and the structure of oxidized PE species containing hydroperoxy-arachidonic acid with assigned fragments (inserts). The fragment with m/z 283.265 [R₁-H] corresponds to stearic acid (C18:0) that is localized in the sn-1 position of PE. Fragment with m/z 319.228 [R₂-H]- corresponding to C20:4-OH and ion with m/z 275.239 formed from hydroxy-arachidonic acid with loss of CO₂ [R₂-CO₂-H]⁻ were observed in the MS² spectrum. The fragment with m/z 480.310 [M-H-R₂CH=C=O]⁻ was generated with the loss of C20:4-OOH as ketene. Fragment m/z 155.142 was produced after C10-C11 bond cleavage and addition of two hydrogens. Fragments attributed to polar head of PE with m/z 140.009 and 196.096 were also formed during MS² fragmentation. Based on fragmentation pattern the species with m/z 782.538 was identified as 15-HETE-PE. **B.** MS/MS spectrum of PE molecular ion with m/z 798.531 at RT = 78.65 min and the structures of oxidized PE species containing hydroperoxy-arachidonic acid with assigned fragments (inserts). The fragment with m/z 283.265 [R₁-H] corresponds to stearic acid (C18:0) that is localized in the sn-1 position of PE. Fragment with m/z 317.211 [R₂-H]- is formed after dehydration of hydroperoxy-arachidonic acid and ion with m/z 273.223 is formed from dehydrated ion with the loss of CO₂ [R₂-H₂O-CO₂-H]⁻. Fragment m/z 113.095 produced after C13-C14 double bond cleavage is indicative of the OOH-group at C15 position of the ketone group formed after loss of water molecule from hydroperoxy-arachidonic acid. Fragments attributed to polar head of PE with m/z 140.0 were also formed during MS² fragmentation. Based on the fragmentation pattern, the species with m/z 798.531 was identified as 15-HpETE-PE.

REVIEWER COMMENTS

Reviewer #1 (Remarks to the Author):

The authors have sufficiently responded to my critiques.

Reviewer #2 (Remarks to the Author):

The authors have adequately and innovatively addressed my comments.

Reviewer #3 (Remarks to the Author):

The manuscript has been revised well. This manuscript would be acceptable if some consideration is given to our concerns below.

Based on your replies (comment1), we understand that "mitophagy is complementarily enhanced under Th2 stress conditions, but this alone is not sufficient to protect mitochondria from oxidative damage". I am not disputing the authors' claims, and I believe that the results of this study are important insights for understanding the redox balance within epithelial cells in the asthma pathogenesis.

Recent research suggests that autophagy is controversial with arguments for a role in both beneficial and detrimental effects in animal models of allergic disease

(1. Theofani E, Autophagy: A Friend or Foe in Allergic Asthma? International Journal of Molecular Sciences. 2021; 22(12):6314., 2. Chen X, et al. International consensus guidelines for the definition, detection, and interpretation of autophagy-dependent ferroptosis. Autophagy. Published online March 24, 2024).

In the context of ferroptosis, various forms of selective autophagy (ferritinophagy, lipophagy, clockphagy ,etc) have been implicated in promoting this form of cell death, usually through indirect mechanisms in which selective autophagy favors biochemical pathways that facilitate pro-ferroptotic oxidative reactions (1, 2). You state in a sentence newly added in this proofreading that "Although mitophagy has been previously reported in severe asthmatic fibroblasts, mitophagy in asthmatic epithelial cells has not been reported."(LINE 480-481), although this is incorrect. There is one report of the involvement of mitophagy in airway epithelial cells (3. Zhang Y, Do DC, Hu X, et

al. CaMKII oxidation regulates cockroach allergen-induced mitophagy in asthma. *J Allergy Clin Immunol.* 2021;147(4) :1464-1477.e11.). This study is contrary to your claims, as it alleges harmful effects of mitophagy in asthma.

On the other hand, there is also support for the contention that, as you allege, under certain circumstances selective autophagy can act as a cytoprotective mechanism during ferroptosis, by selectively removing damaged or dysfunctional cellular components (1,2). Therefore, I would recommend describing that selective or macroautophagy plays a multifaceted role in asthma pathogenesis, with respect to ferroptosis and asthma phenotypes, wherein mitophagy preserves mitochondria and maintains the number and function of cilia in the face of ongoing ferroptosis within the authors' system.

Finally, as this study struggles to understand the mechanism, a Graphical abstract or diagram, if possible, would help to facilitate understanding.

Reviewer #4 (Remarks to the Author):

All of my concerns have been addressed in the revisions.

REVIEWER COMMENTS

Reviewer #1 (Remarks to the Author):

The authors have sufficiently responded to my critiques.

We are pleased that the reviewer feels we have sufficiently responded.

Reviewer #2 (Remarks to the Author):

The authors have adequately and innovatively addressed my comments.

Thank you for that positive comment on our innovative approach!

Reviewer #3 (Remarks to the Author):

The manuscript has been revised well. This manuscript would be acceptable if some consideration is given to our concerns below.

Thank you. We hope that we have addressed your concerns with the following edits to the discussion.

Based on your replies (comment1), we understand that "mitophagy is complementarily enhanced under Th2 stress conditions, but this alone is not sufficient to protect mitochondria from oxidative damage". I am not disputing the authors' claims, and I believe that the results of this study are important insights for understanding the redox balance within epithelial cells in the asthma pathogenesis.

Thank you for your favorable comments regarding the important insights from our studies.

Recent research suggests that autophagy is controversial with arguments for a role in both beneficial and detrimental effects in animal models of allergic disease

(1. Theofani E, Autophagy: A Friend or Foe in Allergic Asthma? International Journal of Molecular Sciences. 2021; 22(12):6314., 2. Chen X, et al. International consensus guidelines for the definition, detection, and interpretation of autophagy-dependent ferroptosis. Autophagy. Published online March 24, 2024).

In the context of ferroptosis, various forms of selective autophagy (ferritinophagy, lipophagy, clockphagy ,etc) have been implicated in promoting this form of cell death, usually through indirect mechanisms in which selective autophagy favors biochemical pathways that facilitate pro-ferroptotic oxidative reactions (1, 2).

We are in complete agreement that the processes we identify could have both positive and negative cell survival effects. Indeed, these are such dynamic processes. We have attempted to address these concerns by adding these reference and several lines to the discussion (see details later in this section).

You state in a sentence newly added in this proofreading that "Although mitophagy has been previously reported in severe asthmatic fibroblasts, mitophagy in asthmatic epithelial cells has not been reported."(LINE 480-481), although this is incorrect. There is one report of the involvement of mitophagy in airway epithelial cells (3. Zhang Y, Do DC, Hu X, et al. CaMKII oxidation regulates cockroach allergen-induced mitophagy in asthma. J Allergy Clin Immunol. 2021;147(4) :1464-1477.e11.). This study is contrary to your claims, as it alleges harmful effects of mitophagy in asthma.

We apologize not including and reference this study in previous, and now have been referenced this in the discussion:

Line 486: and in primary human bronchial epithelial cells (REF 3).

On the other hand, there is also support for the contention that, as you allege, under certain circumstances selective autophagy can act as a cytoprotective mechanism during ferroptosis, by selectively removing damaged or dysfunctional cellular components (1,2). Therefore, I would recommend describing that selective or macroautophagy plays a multifaceted role in asthma pathogenesis, with respect to ferroptosis and asthma phenotypes, wherein mitophagy preserves mitochondria and maintains the number and function of cilia in the face of ongoing ferroptosis within the authors' system.

Thank you very much for your invaluable insight and comments on this. We have tried to incorporate this (and the earlier comment) into the discussion as follows:

Line 485-492: Although mitophagy has been previously reported in severe asthmatic fibroblasts (Ref: Mukhopadhyay S, PLoS One 2020)), and in primary human bronchial epithelial cells (Ref 3), its functional implications in asthma remain controversial (Ref 1,2). Depending on the type of selective autophagy, it can promote cell death, often through indirect mechanisms which favor biochemical pathways that facilitate pro-ferroptotic oxidative reactions (Ref 1,2), or, as we suggest here, act as a cytoprotective mechanism which selectively removes damaged or dysfunctional cellular components. Our results suggest that in the face of ongoing ferroptosis, mitophagy preserves mitochondrial components which may serve to stabilize ciliated cells and their function.

Finally, as this study struggles to understand the mechanism, a Graphical abstract or diagram, if possible, would help to facilitate understanding.

Thank you for this excellent suggestion. A *Graphical abstract* has been included as Figure 7 and embedded with summary discussion paragraph (*Line 505*). If the reviewer or the editor would like this placed in a different location, we are happy to re-arrange as indicated.

*Reviewer #4 (Remarks to the Author):
All of my concerns have been addressed in the revisions.*

Thank you.

REVIEWERS' COMMENTS

Reviewer #3 (Remarks to the Author):

Authors appropriately responded to my comments.